# CONVOLUTION MEETS LORA: PARAMETER EFFICIENT FINETUNING FOR SEGMENT ANYTHING MODEL

**Zihan Zhong**[*]
Tsinghua University
zhongzh22@mails.tsinghua.edu.cn

**Zhiqiang Tang**
Amazon Web Services
zqtang@amazon.com

**Tong He**
Amazon Web Services
htong@amazon.com

**Haoyang Fang**
Amazon Web Services
haoyfang@amazon.com

**Chun Yuan**
Tsinghua University
yuanc@sz.tsinghua.edu.cn

## ABSTRACT

The Segment Anything Model (SAM) stands as a foundational framework for image segmentation. While it exhibits remarkable zero-shot generalization in typical scenarios, its advantage diminishes when applied to specialized domains like medical imagery and remote sensing. To address this limitation, this paper introduces Conv-LoRA, a simple yet effective parameter-efficient fine-tuning approach. By integrating ultra-lightweight convolutional parameters into Low-Rank Adaptation (LoRA), Conv-LoRA can inject image-related inductive biases into the plain ViT encoder, further reinforcing SAM's local prior assumption. Notably, Conv-LoRA not only preserves SAM's extensive segmentation knowledge but also revives its capacity of learning high-level image semantics, which is constrained by SAM's foreground-background segmentation pretraining. Comprehensive experimentation across diverse benchmarks spanning multiple domains underscores Conv-LoRA's superiority in adapting SAM to real-world semantic segmentation tasks.[1]

## 1 INTRODUCTION

The AI community have witnessed the explosion development of a series of foundation models in recent years, such as CLIP (Radford et al., 2021), GPT-4 (OpenAI, 2023) and ViT-22B (Dehghani et al., 2023). Recently, Segment Anything (SAM) (Kirillov et al., 2023), a promptable model pretrained on over 1 billion masks and 11 million images, emerged as a foundation model for image segmentation. Despite its impressive zero-shot performance on generic object segmentation, it doesn't perform well on many real-world segmentation tasks in certain domains (Tang et al., 2023; Ji et al., 2023; Zhou et al., 2023), such as natural images (Borji et al., 2019; Fan et al., 2020a), agriculture (Sriwastwa et al., 2018), remote sensing (Xu et al., 2018) and medical images (Fan et al., 2020b).

Following the pretraining-finetuing paradigm (Dosovitskiy et al., 2020; He et al., 2022; Liu et al., 2021a), it is natural to finetune SAM on downstream tasks to enhance its performance. However, existing works (Zhang & Liu, 2023; Chen et al., 2023; Shaharabany et al., 2023) have failed to either analyze or address certain limitations inherent in SAM. 1) SAM's image encoder is a plain ViT, which is known to lack of vision-specific inductive biases (Chen et al., 2022) that are useful for dense predictions. 2) SAM's pretraining is essentially a binary mask prediction task that, where, given one prompt, it separates foreground object from background. The low-level mask prediction pretraining hinders SAM's ability to capture high-level image semantic information crucial for tasks like multi-class semantic segmentation.

To tackle the above limitations and still retain SAM's valuable segmentation knowledge acquired during pretraining, we finetune a small set of (extra) model parameters while freezing most of SAM's pretrained weights, hence parameter efficient finetuning (PEFT). This raises the question: *Can PEFT*

---

[*]Work done while interning at Amazon Web Services.

[1]Our code is public available at https://github.com/autogluon/autogluon/tree/master/examples/automm/Conv-LoRA

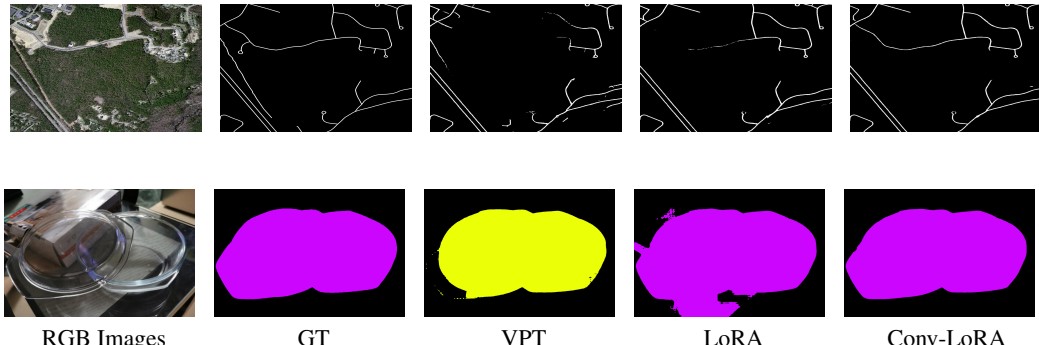

Figure 1: Comparison of VPT, LoRA, and Conv-LoRA (ours) in binary-class road segmentation (**top**) and multi-class transparent object segmentation (**bottom**). Conv-LoRA reinforces image-related local priors, allowing SAM to separate roads from adjacent buildings, while LoRA and VPT struggle in this regard. In the second row, VPT produces a reasonable mask for the bowls but erroneously assigns them to the jar/kettle class (indicated by object color), revealing SAM's limited high-level semantic understanding. Both LoRA and Conv-LoRA rectify this misclassification through finetuing SAM's image encoder, with Conv-LoRA delivering a cleaner mask with fewer boundary artifacts.

*enhance SAM encoder with image-related local prior and facilitate the acquisition of high-level semantic information?*

In this paper, we propose a new PEFT method named Conv-LoRA by diving into Low-Rank Adaptation (LoRA) (Hu et al., 2021). LoRA introduces slim trainable linear projection layers into each transformer layer of SAM's encoder, thereby helping recover its capacity to extract high-level semantic information. Our experiments demonstrate that LoRA surpasses the widely-adopted visual prompt tuning (VPT) (Jia et al., 2022), particularly in the multi-class semantic segmentation tasks. On top of LoRA, Conv-LoRA integrates lightweight convolution layers within its bottleneck structure. Convolution can introduce the image-related local prior (i.e. a pixel exhibits stronger correlation with its neighbors than the distant pixels) (Chen et al., 2022) through the local spatial operations.

Furthermore, it is essential to inject the local prior into the appropriate scale(s) of image features, considering the potential variations in object scales. To this end, Conv-LoRA draws inspiration from the concept of Mixture-of-Experts (MoE) (Shazeer et al., 2017) and incorporates multiple parallel convolutional experts, each specializing in a distinct feature scale. Given that ViT processes image features at a fixed scale, typically downsampling them by a factor of 16 from the original resolution, each expert in Conv-LoRA initially recovers image features at a specific scale, applies convolutional operations, and then reverts the features to the default scale. Compared to ViT-adaptor (Chen et al., 2022) and vision-specific transformers like Swin Transformer (Liu et al., 2021a), Conv-LoRA provides an implicit way to enforce multi-scale local priors, assuming it can leverage image features at the default scale to reconstruct feature information at higher scales. Fortunately, SAM's supervised pretraining, which involves masks of various scales, enables the ViT to acquire knowledge of image features beyond the default scale.

In the spirit of PEFT, we also remove the prompt encoder and add lightweight MLPs in the mask decoder for multi-class prediction. This simple modification has transformed SAM into an end-to-end model that can be finetuned on both binary and multi-class semantic segmentation applications. Overall, our contribution can be summarized as follows:

- We present an innovative PEFT technique Conv-LoRA. By incorporating supplementary convolution operations, Conv-LoRA reinforces the local prior of SAM from the perspective of handling the limitation of plain ViT.

- Conv-LoRA uses MoE to model the process of dynamically selecting the proper feature scale to inject the vision-specific inductive biases.

- Our investigations reveal that SAM's pretraining has impeded its ViT encoder's capacity to learn high-level image semantic information. However, LoRA demonstrates the potential to help SAM recover this crucial ability.

- We conduct an extensive benchmark encompassing diverse domains, including natural images, agriculture, remote sensing, and healthcare. Conv-LoRA consistently exhibits superior performance over other PEFT techniques in various downstream tasks.

## 2   RELATED WORK

**Parameter Efficient Fine-Tuning (PEFT)**. Parameter Efficient Fine-Tuning (PEFT) minimizes computational and storage requirements by selectively fine-tuning a small subset of model parameters, while keeping the majority fixed. PEFT encompasses methods such as adapter-based techniques, selective parameter tuning, prompt-driven fine-tuning, and Low-Rank Adaptation (LoRA) emerging from Natural Language Processing (NLP). In the adapter paradigm (Houlsby et al., 2019; Hu et al., 2021; Sung et al., 2022), compact modules are inserted within transformer layers, and other approaches (Guo et al., 2020; Zaken et al., 2021) involve fine-tuning a small fraction of parameters from pre-trained backbones. Prompt tuning (Lester et al., 2021; Li & Liang, 2021) adds adaptable tokens to input or intermediate sequences, and LoRA (Hu et al., 2021) introduces trainable low-rank matrices into transformer layers for weight updates.

PEFT techniques have also proven effective in the Computer Vision (CV) domain. Visual Prompt Tuning (VPT) (Jia et al., 2022) applies prompt tuning concepts (Lester et al., 2021) to image classification, while Scale and Shift Feature Modulation (SSF) (Lian et al., 2022) uses scale and shift parameters for modulating visual features in image classifiers. Convpass (Jie & Deng, 2022) introduces a convolutional bottleneck to enhance ViT's performance in image classification. In our study, we focus on developing PEFT for SAM in semantic segmentation tasks, specifically enforcing multi-scale local priors beyond the default scale, distinguishing our approach from Convpass.

**Segmentation Models**. FCN (Long et al., 2015) is a key deep image segmentation model that directly generates pixel-wise segmentation maps from images. U-Net (Ronneberger et al., 2015) employs an encoder-decoder structure with skip connections to preserve fine-grained spatial information. Deeplab (Chen et al., 2017a) integrates atrous (dilated) convolutions for multi-scale context, while PSPNet (Zhao et al., 2017) uses a pyramid pooling module. DANet (Fu et al., 2019), SANet (Zhong et al., 2020), and EMA (Li et al., 2019) utilize attention mechanisms for contextual dependencies. Transformer architectures like PVT (Wang et al., 2021), Swin (Liu et al., 2021b), CvT (Wu et al., 2021), CoaT (Xu et al., 2021), LeViT (Graham et al., 2021), Segformer (Xie et al., 2021a), and PVT v2 (Wang et al., 2022) bring various improvements. SAM (Ji et al., 2023), a recent breakthrough in segmentation, offers a universal approach for segmenting diverse objects and regions in images. Fine-tuning SAM on downstream tasks is recommended due to a lack of high-level semantic information and potential domain bias in the pre-training dataset.

**Fine-tuning SAM.** Some prior works (Chen et al., 2023; Zhang & Liu, 2023; Wu et al., 2023; Chai et al., 2023; Shaharabany et al., 2023; Hu et al., 2023; Wang et al., 2023) explore fine-tuning SAM for downstream tasks. These methods include tuning SAM's mask decoder or integrating parameter-efficient tuning methods with SAM's image encoder. Some of them (e.g.,(Chen et al., 2023; Zhang & Liu, 2023; Shaharabany et al., 2023)) provide end-to-end solutions to automate SAM. Our method further addresses the structural limitation of SAM's image encoder for capturing visual-specific inductive biases by introducing convolution operations. And we unveil that SAM's pretraining hampers its ViT encoder's ability to learn high-level semantic information. We also transform SAM into an end-to-end semantic segmentation model with minor architectural adjustments.

**Mixture-of-Experts.** Mixture-of-Expert (MoE) is designed to expand model capacity while introducing small computational overhead. An MoE layer leverages multiple experts to enhance model capacity, while using the gating network to regulate sparsity for computational savings. Feed-Forward Networks (FFN) are commonly employed as the default choice for experts (Shazeer et al., 2017; Riquelme et al., 2021; Bao et al., 2022; Du et al., 2022; Zhou et al., 2022; Fedus et al., 2022). Some efforts (Zuo et al., 2021; Zhou et al., 2022) focus on more efficient gating mechanisms.

In our work, we utilize the concept of MoE, not aiming at improving it. We compare MoE used in our work with original MoE in three aspects: 1) The original goal of MoE is to expand model capacity without excessively increasing computational overhead, whereas ours is to dynamically inject the local prior into the feature maps of different scales. 2) The structures of experts in MoE are typically the same, whereas ours are not. Each expert specializes in a specific scaling operation

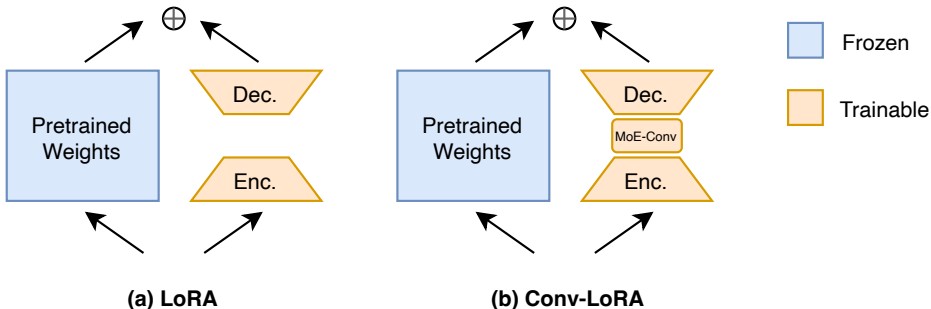

Figure 2: LoRA vs. Conv-LoRA. Both LoRA and Conv-LoRA add an extra trainable encoder-decoder structure parallel to the frozen pre-trained weights. Inside the bottleneck of LoRA, Conv-LoRA inserts lightweight convolution operations managed by MoE with negligible extra parameters.

in our method. 3) While MoE is mostly employed during pre-training, we employ MoE as a part for parameter-efficient tuning on the downstream tasks.

## 3 METHOD

### 3.1 CONV-LORA

**LoRA**. First, let's briefly recap the design of LoRA (Hu et al., 2021), which uses an encoder-decoder structure to impose a low-rank constraint on the weight updates (fig. 2 (a)). It freezes the pre-trained model weights and injects small trainable rank decomposition matrices into each layer of the transformer architecture. Specifically, given a pre-trained weight matrix $W_0 \in \mathbb{R}^{b \times a}$, LoRA adds a pair of linear encoder $W_e$ and decoder $W_d$, i.e., trainable rank decomposition matrices, along its side. $W_e$ and $W_d$ satisfy the low rank constraints $W_e \in \mathbb{R}^{r \times a}$, $W_d \in \mathbb{R}^{b \times r}$, and $r \ll min(a, b)$. With LoRA, the forward pass changes from $h = W_0 x$ to:

$$h = W_0 x + W_d W_e x \tag{1}$$

**Conv-LoRA** aims to incorporate convolution operations between the encoder and decoder components of LoRA (fig. 2 (b)). On one hand, convolution can inject the image-related local prior, addressing fundamental limitation of the vanilla ViT. On the other hand, the low-rank constraint ensures that the convolution layers remain exceedingly lightweight, preserving the PEFT nature of Conv-LoRA.

A pivotal consideration in designing Conv-LoRA is determining the scale of feature maps at which to introduce the local prior. While the feature maps in ViT are uniform in scale, object masks typically encompass a wide range of scales. Therefore, it is crucial to apply convolution operations at the right scale. To tackle this challenge, we draw inspiration from the concept of Mixture of Experts (MoE) (Shazeer et al., 2017). MoE comprises multiple expert networks and a gating module that dynamically selects which expert(s) to activate during the forward pass (fig. 3). Adapting this concept to Conv-LoRA, each expert specializes in convolution at a specific scale of feature maps, and a compact gating module learns to dynamically choose the expert(s) based on the input data. Mathematically, with Conv-LoRA, eq. (1) changes to:

$$h = W_0 x + W_d \left( \sum_{i}^{n} G(W_e x)_i E_i(W_e x) \right) \tag{2}$$

where $W_0 \in \mathbb{R}^{C_{out} \times C_{in}}$, $W_e \in \mathbb{R}^{r \times C_{in}}$, $W_d \in \mathbb{R}^{C_{out} \times r}$, $x \in \mathbb{R}^{B \times C_{in} \times H \times W}$. $B$ is batch size, $C_{in}/C_{out}$ is the number of input / output channels, $H$ and $W$ correspond to the height and width. $E_i$ is the $i_{th}$ expert of all $n$ experts. $G$ is the gating network with only top-k (default 1) values activated. Refer to appendix A for more details of gating.

Inside each expert, three key operations are arranged in sequence: an interpolation function that reconstructs feature maps at a specific scale, a $3 \times 3$ convolutional layer, and a subsequent interpo-

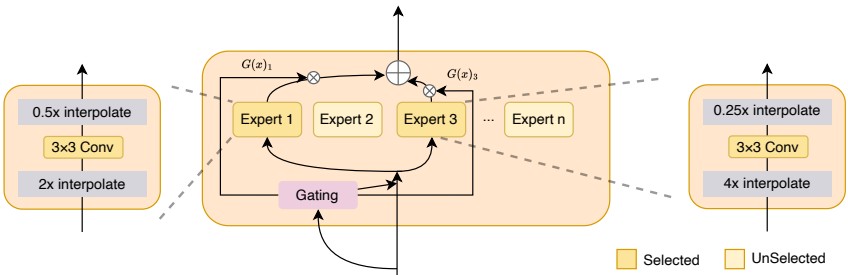

Figure 3: MoE-Conv. It consists of $n$ experts and a gating network for dynamic expert selection. Each expert reconstructs feature maps at a specific scale, applies convolution, and returns the feature maps to the default scale. Each expert specializes in one unique feature scale.

lation operation to map the feature maps back to the default feature scale of ViT. Assume expert $E_i$ is in charge of scale $s_i$, we can formulate it as:

$$E_i(x) = \text{Interpolate}(\text{Conv}_{3\times3}(\text{Interpolate}(x, s_i)), 1/s_i) \tag{3}$$

For instance, if $s_i = 4$, expert $E_i$ would initially upscale the feature maps by a factor of 4x, apply the $\text{Conv}_{3\times3}$ operation, and finally, downscale the feature maps by 4x.

**MoE vs. Multi-scale**. In contrast to MoE, another method to address diverse scales is employing a multi-scale strategy. This approach utilizes multiple branches to concurrently inject local priors at various scales and aggregates the results. Although seemingly more straightforward, this method comes at a higher computational cost when compared to MoE. The efficiency of MoE stems from its capacity to selectively activate sparse experts, thereby minimizing computational overhead. Given our priority on efficient finetuning, we favor MoE as a discerning choice.

## 3.2 END-TO-END MULTI-CLASS SEGMENTATION WITH SAM

SAM comprises three essential components: an image encoder, a prompt encoder, and a mask decoder. When provided with an image and a prompt, which can take the form of a point, box, mask, or text, the mask decoder generates a mask of the object associated with the given prompt. While this prompt-based approach renders SAM flexible for integration into larger systems, such as interactive segmentation or a combination of detection and subsequent segmentation, it does pose challenges in making SAM an end-to-end model in practical applications. To automate SAM, we freeze the prompt encoder, thus always constant prompt tokens to mask decoder, when finetuning it on downstream tasks. Moreover, the original mask decoder is designed to predict binary masks, distinguishing between foreground and background based on the given prompt. To adapt SAM for multi-class semantic segmentation tasks, we introduce a straightforward classification branch (depicted as the red dashed box in fig. 4) within the mask decoder. This extra branch is responsible for predicting classification scores. Additionally, we apply full fine-tuning to the mask decoder as it is a lightweight module. For more comprehensive information, refer to appendix B.

## 4 EXPERIMENTS

**Settings.** We perform several experiments using SAM on four real-world scenarios, including medical images, natural images, agriculture and remote sensing. We use the batch size of 4 and Adam optimizer with learning rate of $1 \times 10^{-4}$ as default, with a weight decay of $1 \times 10^{-4}$. A larger learning rate of $3 \times 10^{-4}$ is found useful for the datasets we use in agriculture and remote sensing. The random horizontal flip is applied during training as data augmentation. All the methods are trained for 30 epochs with structure loss (i.e., the combination of weighted IoU loss and binary cross entropy loss) unless otherwise specified. Additionally, our Conv-LoRA follows Shazeer et al. (2017) to introduce extra loss for balancing the utilization among the experts. The weight of the extra loss is set to 1.0 and 2.0 for binary-class and multi-class semantic segmentation respectively. We set the number of experts to be 8 by default, with each expert specializing in a scaling ratio within the

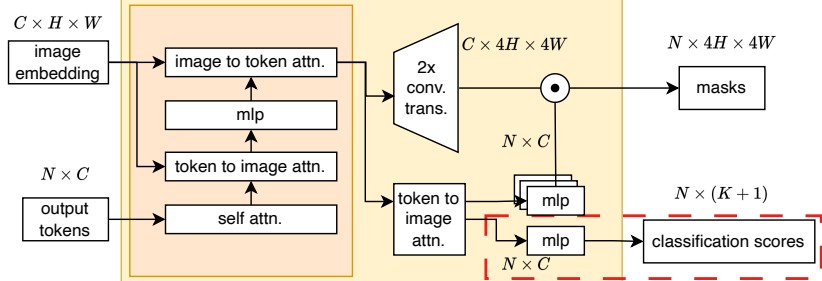

Figure 4: The modified SAM's mask decoder for multi-class semantic segmentation. The classification module (within the red dashed box) is new added compared to original SAM's mask decoder. $N$ is the number of output mask tokens, $K$ is the number of classes, $C$ is the number of channels, $H$ and $W$ indicate the height and width of the feature map (we omit 'batch size' for simplicity).

continuous range from 1 to 8. And we apply Conv-LoRA to the query, key and value matrices in self-attention layers, same as how LoRA does.

**Datasets**. Our experiments encompass semantic segmentation datasets from various domains, spanning natural images, medical images, agriculture, and remote sensing. In the natural image domain, we explore two specific tasks: camouflaged object segmentation (Fan et al., 2020a; Skurowski et al., 2018; Le et al., 2019) and shadow detection (Vicente et al., 2016). Within medical segmentation, we investigate polyp segmentation (Jha et al., 2020; Bernal et al., 2015; Tajbakhsh et al., 2015; Vázquez et al., 2017; Silva et al., 2014) and skin lesion segmentation (Codella et al., 2018). For agriculture and remote sensing, we employ the leaf disease segmentation (Rath, 2023) and road segmentation (Mnih, 2013) datasets as representative examples, respectively. We also explore multi-class transparent object segmentation using Trans10K-v1 (Xie et al., 2020) with 3 classes and Trans10K-v2 (Xie et al., 2021b) with 12 fine-grained classes. Further details about each dataset can be found in appendix C.

**Baselines**. We compare our method with the following methods: 1) Fine-tune SAM's mask decoder only. 2) BitFit (Zaken et al., 2021), which only fine-tunes bias terms in the pre-trained model. 3) Adapter (Houlsby et al., 2019), which inserts the trainable bottleneck layers between the transformer layers. 4) SAM-Adapter (Chen et al., 2023), which further tunes the patch embedded features and learns an extra embedding for high-frequency components for low-level semantic segmentation tasks. And it is one of the pioneer works that apply PEFT method to SAM. 5) VPT (Jia et al., 2022), which inserts learnable tokens to hidden states for each transformer layer. 6) LST (Sung et al., 2022), which inserts a trainable side network parallel to the frozen backbone. To control the number of trainable parameters, the side adapter network employs a pre-trained ViT-Tiny model, similar to SAN (Xu et al., 2023). The features from the frozen backbone and the side adapter network are fused at the global attention layers of SAM. 7) SSF (Lian et al., 2022), which inserts learnable scale and shift parameters to modulate visual features during training. 8) LoRA (Hu et al., 2021) inserts trainable bottleneck layers parallel to the frozen linear weight.

To adhere to the page limit, we select representative datasets of different domains to report the main experiment results. Refer to appendix D for the full experiment results. All the experiments for PEFT methods are run for three times to ease the randomness. The average values and the standard error are reported in table 1.

### 4.1 BINARY-CLASS SEMANTIC SEGMENTATION

In table 1, all PEFT methods consistently outperform fine-tuning the mask decoder alone, underscoring the importance of finetuning the image encoder of SAM. Furthermore, Conv-LoRA surpasses other PEFT techniques across diverse datasets from different domains. Compared to other PEFT methods, Conv-LoRA uses lightweight convolution operations to strengthen the vision-specific local prior, which turns out effective in boosting the segmentation performance. The substantial performance gaps between SAM trained from scratch and Conv-LoRA also underscore the considerable assistance provided by SAM's pretraining knowledge in enhancing downstream task performance.

| Method | #Params (M) / Ratio (%) | Medical | | | | | | Natural Images | | | | Agriculture | | Remote Sensing | |
|---|---|---|---|---|---|---|---|---|---|---|---|---|---|---|---|
| | | Kvasir | | CVC-612 | | ISIC 2017 | | CAMO | | | SBU | Leaf | | Road | |
| | | $S_\alpha\uparrow$ | $E_\phi\uparrow$ | $S_\alpha\uparrow$ | $E_\phi\uparrow$ | Jac↑ | Dice↑ | $S_\alpha\uparrow$ | $E_\phi\uparrow$ | $F_\beta^w\uparrow$ | BER↓ | IoU↑ | Dice↑ | IoU↑ | Dice↑ |
| *Domain Specific* | * / 100% | 90.9 | 94.4 | 92.6 | 95.5 | 80.1 | 87.5 | 80.8 | 85.8 | 73.1 | 3.56 | 62.3 | 74.1 | 59.1 | 73.0 |
| *SAM trained from scratch* | 641.09 / 100% | 78.5 | 82.4 | 85.9 | 91.6 | 73.8 | 82.5 | 61.9 | 67.0 | 40.5 | 5.53 | 52.1 | 65.5 | 55.6 | 71.1 |
| decoder-only | 3.51 / 0.55% | 86.5 | 89.5 | 85.5 | 89.9 | 69.7 | 79.5 | 78.5 | 83.1 | 69.8 | 14.58 | 50.8 | 63.8 | 48.6 | 65.1 |
| BitFit | 3.96 / 0.62% | $90.8 \pm {\scriptstyle .57}$ | $93.8 \pm {\scriptstyle .98}$ | $89.0 \pm {\scriptstyle .40}$ | $91.6 \pm {\scriptstyle .98}$ | $76.4 \pm {\scriptstyle .45}$ | $84.7 \pm {\scriptstyle .35}$ | $86.8 \pm {\scriptstyle .33}$ | $90.7 \pm {\scriptstyle .28}$ | $81.5 \pm {\scriptstyle .19}$ | $3.16 \pm {\scriptstyle .128}$ | $71.4 \pm {\scriptstyle 1.15}$ | $81.7 \pm {\scriptstyle 1.01}$ | $60.6 \pm {\scriptstyle .15}$ | $75.2 \pm {\scriptstyle .11}$ |
| Adapter | 3.92 / 0.61% | $91.2 \pm {\scriptstyle .23}$ | $94.0 \pm {\scriptstyle .16}$ | $89.3 \pm {\scriptstyle .43}$ | $92.0 \pm {\scriptstyle .63}$ | $76.7 \pm {\scriptstyle .66}$ | $85.0 \pm {\scriptstyle .56}$ | $87.7 \pm {\scriptstyle .10}$ | $91.3 \pm {\scriptstyle .40}$ | $82.8 \pm {\scriptstyle .35}$ | $2.84 \pm {\scriptstyle .093}$ | $72.1 \pm {\scriptstyle .47}$ | $82.4 \pm {\scriptstyle .36}$ | $61.5 \pm {\scriptstyle .19}$ | $75.9 \pm {\scriptstyle .12}$ |
| VPT | 4.00 / 0.62% | $91.5 \pm {\scriptstyle .23}$ | $94.3 \pm {\scriptstyle .06}$ | $91.0 \pm {\scriptstyle .04}$ | $93.7 \pm {\scriptstyle 1.41}$ | $76.9 \pm {\scriptstyle .94}$ | $85.1 \pm {\scriptstyle .75}$ | $87.4 \pm {\scriptstyle .60}$ | $91.4 \pm {\scriptstyle .68}$ | $82.1 \pm {\scriptstyle .75}$ | $2.70 \pm {\scriptstyle .055}$ | $73.6 \pm {\scriptstyle .26}$ | $83.8 \pm {\scriptstyle .26}$ | $60.2 \pm {\scriptstyle 1.87}$ | $74.9 \pm {\scriptstyle 1.50}$ |
| LST | 11.49 / 1.77% | $89.7 \pm {\scriptstyle .25}$ | $93.3 \pm {\scriptstyle .37}$ | $89.4 \pm {\scriptstyle .37}$ | $92.4 \pm {\scriptstyle .54}$ | $76.4 \pm {\scriptstyle 1.05}$ | $84.9 \pm {\scriptstyle .79}$ | $83.3 \pm {\scriptstyle .28}$ | $88.0 \pm {\scriptstyle .23}$ | $77.1 \pm {\scriptstyle .02}$ | $3.18 \pm {\scriptstyle .012}$ | $70.2 \pm {\scriptstyle .87}$ | $81.1 \pm {\scriptstyle .82}$ | $60.2 \pm {\scriptstyle .26}$ | $74.9 \pm {\scriptstyle .22}$ |
| SAM-Adapter | 3.98 / 0.62% | $89.6 \pm {\scriptstyle .24}$ | $92.5 \pm {\scriptstyle .10}$ | $89.6 \pm {\scriptstyle .22}$ | $92.4 \pm {\scriptstyle 1.06}$ | $76.1 \pm {\scriptstyle .45}$ | $84.6 \pm {\scriptstyle .37}$ | $85.6 \pm {\scriptstyle .26}$ | $89.6 \pm {\scriptstyle .55}$ | $79.8 \pm {\scriptstyle .89}$ | $3.14 \pm {\scriptstyle .063}$ | $71.4 \pm {\scriptstyle .20}$ | $82.1 \pm {\scriptstyle .10}$ | $60.6 \pm {\scriptstyle .06}$ | $75.2 \pm {\scriptstyle .04}$ |
| SSF | 4.42 / 0.69% | $91.3 \pm {\scriptstyle .87}$ | $93.9 \pm {\scriptstyle 1.49}$ | $89.6 \pm {\scriptstyle .37}$ | $91.9 \pm {\scriptstyle .79}$ | $76.6 \pm {\scriptstyle .19}$ | $85.0 \pm {\scriptstyle .14}$ | $87.5 \pm {\scriptstyle .11}$ | $91.4 \pm {\scriptstyle .16}$ | $82.6 \pm {\scriptstyle .12}$ | $3.19 \pm {\scriptstyle .046}$ | $71.5 \pm {\scriptstyle .63}$ | $81.8 \pm {\scriptstyle .44}$ | $61.6 \pm {\scriptstyle .03}$ | $76.0 \pm {\scriptstyle .02}$ |
| LoRA | 4.00 / 0.62% | $91.2 \pm {\scriptstyle .28}$ | $93.8 \pm {\scriptstyle .22}$ | $90.7 \pm {\scriptstyle .04}$ | $92.5 \pm {\scriptstyle .41}$ | $76.6 \pm {\scriptstyle .23}$ | $84.9 \pm {\scriptstyle .22}$ | $88.0 \pm {\scriptstyle .24}$ | $91.9 \pm {\scriptstyle .42}$ | $82.8 \pm {\scriptstyle .16}$ | $2.74 \pm {\scriptstyle .079}$ | $73.7 \pm {\scriptstyle .20}$ | $83.6 \pm {\scriptstyle .13}$ | $62.2 \pm {\scriptstyle .21}$ | $76.5 \pm {\scriptstyle .18}$ |
| Conv-LoRA | 4.02 / 0.63% | $\mathbf{92.0} \pm {\scriptstyle .15}$ | $\mathbf{94.7} \pm {\scriptstyle .16}$ | $\mathbf{91.3} \pm {\scriptstyle .69}$ | $\mathbf{94.0} \pm {\scriptstyle .78}$ | $\mathbf{77.6} \pm {\scriptstyle .57}$ | $\mathbf{85.7} \pm {\scriptstyle .36}$ | $\mathbf{88.3} \pm {\scriptstyle .60}$ | $\mathbf{92.4} \pm {\scriptstyle .31}$ | $\mathbf{84.0} \pm {\scriptstyle .34}$ | $\mathbf{2.54} \pm {\scriptstyle .081}$ | $\mathbf{74.5} \pm {\scriptstyle .39}$ | $\mathbf{84.3} \pm {\scriptstyle .34}$ | $\mathbf{62.6} \pm {\scriptstyle .36}$ | $\mathbf{76.8} \pm {\scriptstyle .27}$ |

Table 1: Results on binary semantic segmentation. '# Params (M) / Ratio (%)' represents the number of trainable parameters and its proportion relative to the total number. *Domain Specific* is a placeholder referring to methods that specifically designed for the tasks. 'Underlined' denotes the better results compared to PEFT methods. See appendix D for more details. Compared to LoRA, Conv-LoRA incurs negligible parameter overhead, but delivers a clear performance boost.

| Method | #Params (M) / Ratio (%) | Easy | | | | Hard | | | |
|---|---|---|---|---|---|---|---|---|---|
| | | Acc↑ | mIoU↑ | MAE↓ | MBER↓ | Acc↑ | mIoU↑ | MAE↓ | MBER↓ |
| *TransLab* | 42.19 / 100% | 95.77 | 92.23 | 0.036 | 3.12 | 83.04 | 72.10 | 0.166 | 13.30 |
| decoder-only | 3.51 / 0.55% | 94.68 | 88.54 | 0.050 | 4.24 | 83.53 | 68.30 | 0.186 | 14.37 |
| VPT | 4.00 / 0.62% | 98.31 | 95.73 | 0.017 | 1.52 | 90.42 | 83.38 | 0.083 | 7.21 |
| LoRA | 4.00 / 0.62% | 98.44 | 96.26 | 0.016 | 1.35 | 91.94 | 83.95 | 0.083 | 6.35 |
| Conv-LoRA | 4.02 / 0.63% | **98.63** | **96.45** | **0.015** | **1.27** | **93.05** | **84.37** | **0.075** | **6.25** |

| Method | # Params (M) / Ratio (%) | Acc↑ | mIoU↑ | Category IoU↑ | | | | | | | | | | | |
|---|---|---|---|---|---|---|---|---|---|---|---|---|---|---|---|
| | | | | bg | shelf | jar | freezer | window | door | eyeglass | cup | wall | bowl | bottle | box |
| *TransLab* | 42.19 / 100% | 92.67 | 69.00 | 93.90 | 54.36 | 64.48 | 65.14 | 54.58 | 57.72 | 79.85 | 81.61 | 72.82 | 69.63 | 77.50 | 56.43 |
| *Trans2Seg* | 56.20 / 100% | 94.14 | 72.15 | 95.35 | 53.43 | 67.82 | 64.20 | 59.64 | 60.56 | 88.52 | 86.67 | 75.99 | 73.98 | 82.43 | 57.17 |
| decoder-only | 3.51 / 0.55% | 90.66 | 49.97 | 93.66 | 32.75 | 39.96 | 35.87 | 50.70 | 45.89 | 57.38 | 73.16 | 69.36 | 54.23 | 56.58 | 33.77 |
| VPT | 4.00 / 0.62% | 94.42 | 62.81 | 97.41 | 29.76 | 52.82 | 62.09 | 55.54 | 63.61 | 81.12 | 83.40 | 79.61 | 65.29 | 72.92 | 44.77 |
| LoRA | 4.00 / 0.62% | 94.80 | 66.01 | 97.50 | 42.17 | 57.82 | **64.35** | 53.44 | 64.08 | **87.28** | **85.28** | 80.43 | 63.67 | 77.97 | 49.56 |
| Conv-LoRA | 4.02 / 0.63% | **95.07** | **67.09** | **97.66** | **50.51** | **58.44** | 51.70 | **55.69** | **65.22** | 85.23 | 84.84 | **80.97** | **72.84** | **79.83** | **52.73** |

Table 2: Results on multi-class semantic segmentation. The tables are the results for three-class and twelve-class semantic segmentation respectively.

Additionally, while Conv-LoRA outperforms certain domain-specific methods that are having more trainable parameters, it may still fall short on specific datasets. It's important to note that Conv-LoRA aims to be a general-purpose PEFT method for adapting SAM to various domains rather than competing with these domain-specific models. Tailoring SAM for specific domain applications with more intricate adjustments might yield superior performance compared to specialized model designs.

## 4.2 MULTI-CLASS SEMANTIC SEGMENTATION

In table 2, while decoder-only fine-tuning approaches achieve comparable segmentation accuracy with domain-specific methods, they exhibit a substantial gap in terms of mIoU metrics. While accuracy measures pixel-level segmentation performance, mIoU takes mask class information into account. We suspect that SAM's image encoder encounters challenges in extracting high-level semantic information that is valuable for classification tasks. Moreover, fine-tuning the image encoder using PEFT methods results in a significant boost in mIoU, indicating a restoration of its capability to learn high-level image semantics.

Additionally, we conduct linear probing experiments on SAM's image encoder. Specifically, we freeze SAM's ViT-B encoder and train only a linear head on ImageNet-1K. Similarly, we perform linear probing for the ViT-B pretrained with MAE (He et al., 2022). The results reveal that SAM's image encoder exhibits significantly lower ImageNet-1K accuracy compared to the MAE encoder (54.2% vs. 67.7%). Given that SAM's image encoder is initialized using an MAE encoder, we

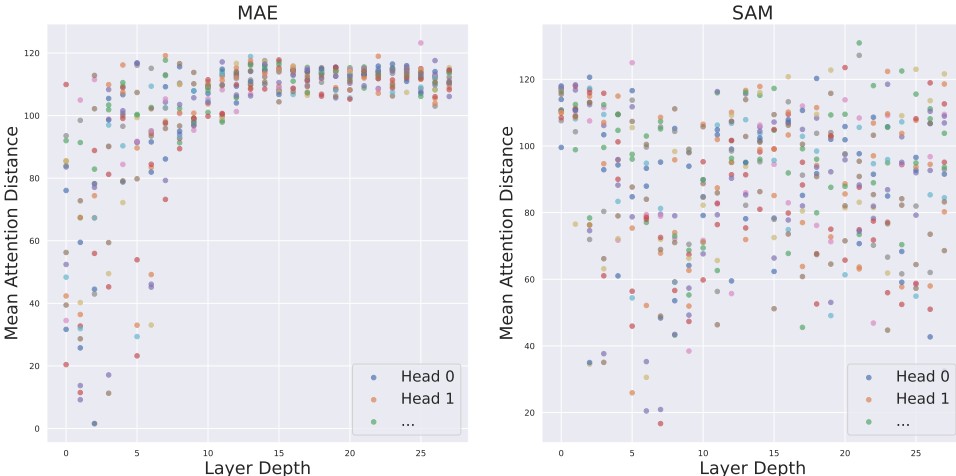

Figure 5: Mean attention distance of each attention head, with each dot indicating the mean distance across images for one of the 16 heads at one layer. In contrast to MAE, SAM retains the ability to incorporate local information even in deeper layers.

hypothesize the pre-training focused on low-level foreground-background mask prediction adversely affects the ViT encoder's ability to capture high-level semantic information for classification.

To be specific, SAM was trained on a dataset that exclusively comprises segmentation masks without explicit semantic information. In theory, to minimize loss, the fundamental objective for its encoder is to project pixels into a metric space where pixels from the same object are in close proximity, while those from distinct objects are distantly positioned. This projection requires an implicit understanding of 'objectness', focusing on proximity within an image rather than preserving consistent representations of the same-type object across different images. This introduces a potential challenge in aligning representations with semantics across diverse images.

## 4.3 ABLATION STUDY

**SAM's local prior assumption.** SAM's local prior is grounded in its extensive segmentation pretraining. Through supervised training on a vast dataset encompassing 1 billion high-quality masks and 11 million images, **SAM has honed a robust capability to discern and capture local features within images**. Notably, SAM's encoder retains the ViT architecture, which inherently lacks a dedicated local prior. However, **this deficiency is effectively compensated by the significant local prior acquired through segmentation pretraining.**

We analyze using the mean attention distance as a metric. In fig. 5, SAM exhibits numerous heads in the deep blocks with short mean attention distances, suggesting its heightened focus on local information during the later stages of the encoder. In contrast, the MAE pretrained ViT, representing SAM's initialization, displays consistently long mean attention distances among attention heads in the later stages. Consequently, SAM's segmentation pretraining induces a transformative shift in ViT's attention heads, steering them from a global-oriented to a local-oriented configuration. **This transformation underscores the efficacy of SAM's approach in imbuing the model with a distinctive local prior, enhancing its ability to capture fine-grained details within images**.

**MoE vs. Multi-scale.** Conv-LoRA leverages MoE to dynamically inject the local prior into feature maps of different scales (fig. 6 (a)), as the optimal scale remains unclear. Here we explore the impact of a multi-scale strategy, which fuses the features from all scales simultaneously (fig. 6 (b)).

We compare performance and training costs in table 3. Dynamic MoE outperforms multi-scale direct addition in terms of performance. This could be attributed to the preference for specific scales' feature maps based on different inputs. When injecting the local prior into feature maps of all scales simultaneously, the discrepancy of the importance diminishes as the information from critical feature maps is smoothed out. However, with dynamic selection of the top-1 experts for each forward pass, the information from these crucial scales takes precedence and isn't diluted by other

scales. Dynamic MoE also provides a 1.54x speedup and reduces memory usage by 1.7GB during training. In summary, the comparison underscores the effectiveness and efficiency of MoE.

| Method | ISIC 2017 | | | |
|---|---|---|---|---|
| | Jac ↑ | T-Jac ↑ | Dice ↑ | Acc ↑ |
| Multi-scale | 77.4 | 69.5 | 85.4 | 93.7 |
| MoE | **77.9** | **70.3** | **85.9** | **93.9** |

| Method | Training Speed (Iter / s) ↑ | Training Memory (GB) ↓ |
|---|---|---|
| Multi-scale | 0.79 | 23.4 |
| MoE | **1.22** | **21.7** |

Table 3: MoE vs. Multi-scale, the latter fuses features from all scales simultaneously. The performance and the training cost illustrate the effectiveness and efficiency of dynamic selecting the experts, i.e., injecting the local prior into the feature maps of different scales dynamically.

**The 'Optimal' Scale of Feature Map for Different Datasets.** While we are unable to definitively determine the optimal scale for introducing the local prior, we could check whether the 'optimal' scale within a given range varies indeed across different datasets.

Specifically, we simply modify Conv-LoRA: use only one expert and a specific scaling ratio. We set the scaling ratio to 1, 2, 4 respectively. The experiments are conducted on Leaf Disease Segmentation dataset and ISIC 2017 dataset.

| Scaling Ratio | Leaf | | | ISIC 2017 | | | |
|---|---|---|---|---|---|---|---|
| | IoU ↑ | Dice ↑ | Acc ↑ | Jac ↑ | T-Jac ↑ | Dice ↑ | Acc ↑ |
| 1 | 73.6 | 83.4 | 95.5 | 76.8 | 69.1 | 85.0 | 93.7 |
| 2 | 73.8 | 83.7 | 95.8 | **77.3** | **69.7** | **85.4** | **93.8** |
| 4 | **74.0** | **83.7** | 95.9 | 76.9 | 68.4 | 85.1 | 93.7 |
| 8 | 73.2 | 83.1 | **96.0** | 76.9 | 68.3 | 85.2 | 93.6 |

Table 4: Comparison of the 'optimal' scale of feature map for local prior injection across different datasets. We modify Conv-LoRA to use one expert with a specific scaling ratio.

In table 4, the 'optimal' scale indeed varies across the different datasets. The scaling ratio set to 4 is optimal for Leaf Disease Segmentation dataset, whereas it is set to 2 for ISIC 2017 dataset. These results further confirms our assumption and the necessity for our dynamic local prior injection based on different inputs.

For more ablation experiments, analyses (e.g., further analyses of local prior and MoE) and visualization, refer to appendix E through appendix I.

## 5 CONCLUSION

Parameter efficient finetuning (PEFT) is a popular way when adapting foundation models to various downstream tasks. We present Conv-LoRA, a novel PEFT approach for applying SAM to downstream segmentation applications. Conv-LoRA is simple, generic, and obtains promising results over multiple domains including natural images, agriculture, remote sensing, and healthcare. Moreover, ours shed light on several aspects of SAM: 1) although the large-scale supervised segmentation pretraining can provide image-related local prior knowledge from the data perspective, injecting lightweight convolution operations in the ViT encoder can further boost the exploitation of local prior from another perspective of architecture; 2) the foreground-background segmentation pretraining prevents the image encoder from learning high-level semantic information, which can be alleviated through finetuning relatively few parameters in the encoder.

Our efforts primarily focus on developing a general PEFT method for SAM, showing stronger performance than existing PEFT methods in a broad spectrum of benchmarks, other than directly competing with state-of-the-art (SOTA) models in specialized domains. Given that SAM fine-tuned with Conv-LoRA may not yet consistently outperform domain-specific SOTA models, we believe that tailoring the mask decoder and prompt encoder beyond image encoder finetuning, and combining Conv-LoRA with other PEFT methods can be promising directions for domain-specific applications.

ACKNOWLEDGEMENTS

We thank all the anonymous reviewers for their helpful comments. This work was supported by the National Key R&D Program of China (2022YFB4701400/4701402), SSTIC Grant (KJZD20230923115106012), Shenzhen Key Laboratory (ZDSYS20210623092001004), and Beijing Key Lab of Networked Multimedia.

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

## A  GATING NETWORK OF CONV-LORA

In order to calculate the gating scores $G(x) \in \mathbb{R}^{B \times k}$, i.e., the scores for $B$ samples in a batch assigned to $k$ experts, we need to calculate the values $H(x) \in \mathbb{R}^{B \times n}$ of $n$ experts first. As the size of input $x$ is $B \times r \times H \times W$ (we use $x$ to represent $W_e x$ in eq. (2) for simplicity), we apply global average pooling (denoted as 'AvgPool') followed by a reshaping operation (denoted as 'Reshape'), to obtain $x_h \in \mathbb{R}^{B \times r}$. Specifically, the size of $x$ is changed to $B \times r \times 1 \times 1$ after 'AvgPool', and then changed to $B \times r$ after 'Reshape'. Then, following (Shazeer et al., 2017), we apply the trainable gating weight $W_g \in \mathbb{R}^{r \times n}$, and the noise term $W_{noise} \in \mathbb{R}^{r \times n}$ to calculate $H(x)$:

$$x_h = \text{Reshape}(\text{AvgPool}(x), (B, r))$$
$$H(x)_i = (x_h \cdot W_g)_i + \text{StandardNormal}() \cdot \text{Softplus}((x_h \cdot W_{noise})_i) \tag{4}$$

We keep only the top k values (denoted as 'KeepTopK') based on $H(x)_i$ of each expert $E_i$, setting the rest to $-\infty$, whose corresponding gate values is equal to 0 after a 'Softmax' operation (If $G(x)_j$ is 0, we need not compute $E_j(x)$):

$$G(x) = \text{Softmax}(\text{KeepTopK}(H(x), k))$$
$$\text{where KeepTopK}(v, k)_i = \begin{cases} v_i & \text{if } v_i \text{ is in the top } k \text{ elements of } v. \\ -\infty & \text{otherwise.} \end{cases} \tag{5}$$

## B  MASK DECODER FOR MULTI-CLASS SEGMENTATION

SAM's mask decoder follows the design in mask-based segmentation, wherein the image is grouped into $N$ (i.e., the number of output tokens in fig. 4) regions represented by binary masks. However, unlike other mask-based methods such as Maskformer (Cheng et al., 2021) and Mask2former (Cheng et al., 2022), SAM does not incorporate a classification module; it solely identifies the foreground, lacking the ability for multi-class segmentation. What we need to do is straightforward: incorporate a classification branch to obtain class predictions for the corresponding $N$ masks.

During training, we need to match the set of $N$ mask predictions and the set of ground truth segments. We follow the design in Mask2Former (Cheng et al., 2022), which proposes a memory-friendly way for bipartite matching between the predictions and ground truths. It efficiently alleviates the memory demands in semantic segmentation tasks that necessitate high-resolution mask prediction, particularly for SAM involving the processing of high-resolution input images ($1024 \times 1024$). For the masks that are not allocated to any ground truths, an extra 'no object' category is introduced to ensure the one-to-one matching. Hence, the classification branch should produce $K + 1$ class predictions, assuming there are originally $K$ categories. For semantic inference, we can exclude the 'no object' category and perform a straightforward matrix multiplication between the masks and the classification predictions to obtain pixel-wise predictions.

Following Mask2Former, we use binary cross-entropy loss $\mathcal{L}_{ce}$ and dice loss $\mathcal{L}_{dice}$ for mask loss, cross entropy loss $\mathcal{L}_{cls}$ for classification predictions:

$$\mathcal{L}_{multi-class} = \lambda_{mask}\mathcal{L}_{mask} + \lambda_{cls}\mathcal{L}_{cls} + \lambda_{MoE}\mathcal{L}_{MoE} \tag{6}$$

where $\mathcal{L}_{mask} = \lambda_{ce}\mathcal{L}_{ce} + \lambda_{dice}\mathcal{L}_{dice}$.

## C  DATASET DETAILS

**Polyp Segmentation** (*medical images*) Polyp Segmentation, the task to identify abnormal growths known as polyps within gastrointestinal endoscopic images, plays a critical role in early colorectal cancer diagnosis and treatment planning, and it presents a formidable challenge due to the considerable diversity in polyp shapes and sizes. We choose two polyp segmentation datasets: Kvasir (Jha

et al., 2020) and CVC-ClinicDB/CVC-612 (Bernal et al., 2015). Kvasir contains 1000 images and CVC-ClinicDB, also called CVC-612, includes 612 open-access images. Fan et al. (2020b) divides the images into a 9:1 ratio for training and testing. Additionally, we randomly divide a validation set comprising 20% of the images from the training set, for validation during training. Furthermore, we also use images from CVC-ColonDB (Tajbakhsh et al., 2015), EndoScene (Vázquez et al., 2017) and ETIS (Silva et al., 2014) for testing following the setting in (Fan et al., 2020b). We consistently train all methods for 30 epochs, unless otherwise specified for specific datasets.

**Skin Lesion Segmentation** (*medical images*) Skin Lesion Segmentation involves the segmentation of various types of skin lesions within medical images, serving a crucial role in the early diagnosis and treatment of skin disorders, notably skin cancer. However, this task is particularly challenging due to ambiguous boundaries and color variations. We choose ISIC 2017 (Codella et al., 2018) for skin lesion segmentation. ISIC 2017 provides 2000 images for training, 150 images for validation and 600 images for testing.

**Camouflaged Object Segmentation** (*natural images*) Camouflaged Object Segmentation focuses on identifying objects that are concealed within complex or visually cluttered backgrounds, which is more challenging comparing to traditional object segmentation. We choose three camouflaged object detection datasets: COD10K (Fan et al., 2020a), CHAMELEON (Skurowski et al., 2018), and CAMO (Le et al., 2019). COD10K contains 3040 training and 2026 testing samples. CHAMELEON includes 76 images collected from the Internet for testing. And CAMO provides 1000 images for training and 250 for testing. Following (Fan et al., 2020a), we train on the combined dataset consists of the training images from COD10K and CAMO for 20 epochs, and test on the three datasets. Addtionally, we randomly split 10% of the images from the training set for validation.

**Shadow Detection** (*natural images*) Shadow Detection focuses on the recognition of shadow regions within a scene and can facilitate the estimation of lighting conditions or the removal of shadows. We choose SBU (Vicente et al., 2016), which is the largest annotated shadow dataset. SBU contains 4085 and 638 images for training and testing. We randomly split 10% of the images from the training set for validation and we train the methods for 10 epochs with balanced binary cross entropy loss.

**Leaf Segmentation** (*agriculture*) Leaf segmentation involves the identification of individual plant leaves within agricultural images and plays a crucial role in advancing automation for plant diseases control and high quality food production. We choose a Leaf Disease Segmentation dataset (Rath, 2023), which contains 498 images for training and 90 images for testing. We randomly split the training images into 80% for training, 20% for validation.

**Road Segmentation** (*remote sensing*) Road segmentation detects road or street regions within images or video frames, and is crucial for applications in autonomous driving, traffic analysis, and urban planning. We choose Massachusetts Roads Dataset (Mnih, 2013), which contains 1107 images for training, 13 images for validation and 48 images for testing. And we train the methods for 20 epochs.

**Multiclass Semantic Segmentation** (*natural images*) We choose Trans10K-v1 (Xie et al., 2020) and Trans10K-v2 (Xie et al., 2021b) dataset for multi-class transparent object segmentation. Trans10K-v1 dataset contains 10428 images, with background as one category and two more categories of transparent objects: Transparent Things (e.g., cups, bottles) and Stuff (e.g., windows). Trans10K-v2 dataset is based on Trans10K-v1 dataset, with more fine-grained categories annotations. The dataset contains background plus two main categories divided into 11 fine-grained categories: 1) Transparent Things containing cup, bottle, jar, bowl and eyeglass. 2) Transparent Stuff containing windows, shelf, box, freezer, glass walls and glass doors. In respect to fine-grained categories and high diversity, Trans10K-v2 is more challenging than Trans10K-v1. All the datasets use 5000, 1000 and 4428 images for training, validation and testing, respectively.

# D    FULL EXPERIMENT RESULTS

Here are the full experiment results for binary semantic segmentation. Our Conv-LoRA demonstrates superiority across diverse datasets from various domains compared to other PEFT methods. Noted that models indicated in *italics* in the following tables are specifically designed for the corresponding tasks. We re-run the experiments on all the datasets using the model code provided by

the authors. We choose methods that are either widely established or relatively novel within their respective domains. With the exception of the Leaf Segmentation dataset, which is sourced from Kaggle, we utilize the author's provided sample code to conduct our experiments.

**Polyp Segmentation**:

| Method | # Params (M) / Ratio(%) | Kvasir | | | | CVC-612 | | | |
|---|---|---|---|---|---|---|---|---|---|
| | | $S_\alpha \uparrow$ | $E_\phi \uparrow$ | $F_\beta^\omega \uparrow$ | MAE $\downarrow$ | $S_\alpha \uparrow$ | $E_\phi \uparrow$ | $F_\beta^\omega \uparrow$ | MAE $\downarrow$ |
| *PraNet (Fan et al., 2020b)* | 32.55 / 100% | 90.9 | 94.4 | 88.7 | 2.9 | 92.6 | 95.5 | 88.7 | 1.0 |
| decoder-only | 3.51 / 0.55% | 86.5 | 89.5 | 77.9 | 5.1 | 85.5 | 89.9 | 74.7 | 3.0 |
| BitFit | 3.96 / 0.62% | $90.8 \pm 0.57$ | $93.8 \pm 0.98$ | $87.4 \pm 0.50$ | $3.2 \pm 0.43$ | $89.0 \pm 0.40$ | $91.6 \pm 0.98$ | $81.7 \pm 0.39$ | $2.3 \pm 0.15$ |
| Adapter | 3.92 / 0.61% | $91.2 \pm 0.23$ | $94.0 \pm 0.16$ | $88.2 \pm 1.28$ | $3.1 \pm 0.06$ | $89.3 \pm 0.43$ | $92.0 \pm 0.63$ | $82.5 \pm 0.41$ | $2.2 \pm 0.10$ |
| VPT | 4.00 / 0.62% | $91.5 \pm 0.23$ | $94.3 \pm 0.06$ | $90.0 \pm 0.54$ | $2.8 \pm 0.07$ | $91.0 \pm 0.94$ | $93.7 \pm 1.41$ | $84.8 \pm 2.16$ | $2.1 \pm 0.28$ |
| LST | 11.49 / 1.77% | $89.7 \pm 0.25$ | $93.3 \pm 0.37$ | $86.9 \pm 0.28$ | $3.7 \pm 0.10$ | $89.4 \pm 0.37$ | $92.4 \pm 0.54$ | $83.7 \pm 0.59$ | $2.4 \pm 0.17$ |
| SAM-Adapter | 3.98 / 0.62% | $89.6 \pm 0.24$ | $92.5 \pm 0.10$ | $86.9 \pm 0.31$ | $3.6 \pm 0.09$ | $89.6 \pm 0.22$ | $92.4 \pm 1.06$ | $82.2 \pm 0.51$ | $2.2 \pm 0.07$ |
| SSF | 4.42 / 0.69% | $91.3 \pm 0.87$ | $93.9 \pm 1.49$ | $88.3 \pm 1.42$ | $3.0 \pm 0.57$ | $89.6 \pm 0.37$ | $91.9 \pm 0.79$ | $82.0 \pm 0.74$ | $2.2 \pm 0.18$ |
| LoRA | 4.00 / 0.62% | $91.2 \pm 0.28$ | $93.8 \pm 0.22$ | $88.4 \pm 0.46$ | $3.1 \pm 0.17$ | $90.7 \pm 0.04$ | $92.5 \pm 0.41$ | $84.5 \pm 1.03$ | $2.2 \pm 0.19$ |
| Conv-LoRA | 4.02 / 0.63% | $\mathbf{92.0} \pm 0.15$ | $\mathbf{94.7} \pm 0.16$ | $\mathbf{89.7} \pm 0.60$ | $\mathbf{2.6} \pm 0.06$ | $\mathbf{91.3} \pm 0.69$ | $\mathbf{94.0} \pm 0.78$ | $\mathbf{85.5} \pm 0.97$ | $\mathbf{1.9} \pm 0.17$ |

| Method | # Params (M) / Ratio(%) | CVC-ColonDB | | | | ETIS | | | | CVC-T | | | |
|---|---|---|---|---|---|---|---|---|---|---|---|---|---|
| | | $S_\alpha \uparrow$ | $E_\phi \uparrow$ | $F_\beta^\omega \uparrow$ | MAE $\downarrow$ | $S_\alpha \uparrow$ | $E_\phi \uparrow$ | $F_\beta^\omega \uparrow$ | MAE $\downarrow$ | $S_\alpha \uparrow$ | $E_\phi \uparrow$ | $F_\beta^\omega \uparrow$ | MAE $\downarrow$ |
| *PraNet (Fan et al., 2020b)* | 32.55 / 100% | 82.0 | 84.5 | 70.9 | 4.0 | 79.3 | 80.6 | 58.2 | 2.3 | 93.9 | 97.1 | 85.2 | 0.8 |
| decoder-only | 3.51 / 0.55% | 76.7 | 80.7 | 59.5 | 5.2 | 67.9 | 71.4 | 41.0 | 7.4 | 86.4 | 88.4 | 67.5 | 2.3 |
| BitFit | 3.96 / 0.62% | $83.8 \pm 0.25$ | $86.8 \pm 0.38$ | $72.7 \pm 0.19$ | $3.9 \pm 0.12$ | $84.7 \pm 0.37$ | $87.1 \pm 0.73$ | $67.4 \pm 1.37$ | $1.7 \pm 0.13$ | $91.5 \pm 0.55$ | $94.2 \pm 0.18$ | $81.3 \pm 1.86$ | $1.4 \pm 0.12$ |
| Adapter | 3.92 / 0.61% | $83.6 \pm 0.24$ | $86.3 \pm 0.32$ | $71.7 \pm 0.37$ | $3.6 \pm 0.24$ | $85.3 \pm 0.64$ | $86.9 \pm 0.26$ | $67.0 \pm 0.98$ | $1.8 \pm 0.26$ | $92.9 \pm 0.19$ | $94.6 \pm 0.56$ | $84.1 \pm 0.76$ | $1.2 \pm 0.15$ |
| VPT | 4.00 / 0.62% | $83.9 \pm 0.23$ | $87.3 \pm 0.54$ | $72.9 \pm 0.47$ | $3.5 \pm 0.02$ | $86.3 \pm 0.51$ | $88.0 \pm 0.0$ | $69.4 \pm 0.0$ | $1.8 \pm 0.09$ | $\mathbf{94.6} \pm 0.14$ | $\mathbf{97.7} \pm 0.04$ | $\mathbf{88.2} \pm 0.57$ | $\mathbf{0.6} \pm 0.01$ |
| LST | 11.49 / 1.77% | $82.5 \pm 0.23$ | $86.6 \pm 0.41$ | $72.2 \pm 0.65$ | $4.3 \pm 0.09$ | $81.5 \pm 0.88$ | $84.3 \pm 1.03$ | $62.2 \pm 2.04$ | $3.4 \pm 0.28$ | $92.0 \pm 0.49$ | $93.7 \pm 0.78$ | $82.5 \pm 0.75$ | $1.2 \pm 0.16$ |
| SAM-Adapter | 3.98 / 0.62% | $83.1 \pm 0.66$ | $86.3 \pm 0.50$ | $70.8 \pm 1.15$ | $3.8 \pm 0.21$ | $83.2 \pm 0.75$ | $85.3 \pm 1.75$ | $63.5 \pm 1.36$ | $2.2 \pm 0.41$ | $92.1 \pm 0.28$ | $94.2 \pm 0.91$ | $81.8 \pm 0.52$ | $1.2 \pm 0.20$ |
| SSF | 4.42 / 0.69% | $83.9 \pm 0.15$ | $86.9 \pm 0.33$ | $72.1 \pm 0.72$ | $3.9 \pm 0.10$ | $84.7 \pm 0.42$ | $87.4 \pm 0.48$ | $66.7 \pm 0.65$ | $1.7 \pm 0.05$ | $92.1 \pm 0.07$ | $93.9 \pm 0.66$ | $83.6 \pm 0.24$ | $1.4 \pm 0.23$ |
| LoRA | 4.00 / 0.62% | $84.4 \pm 0.26$ | $87.2 \pm 0.04$ | $73.5 \pm 0.67$ | $4.1 \pm 0.35$ | $85.5 \pm 0.59$ | $86.5 \pm 0.83$ | $68.4 \pm 1.48$ | $1.8 \pm 0.37$ | $93.5 \pm 0.33$ | $95.9 \pm 0.54$ | $85.7 \pm 1.33$ | $0.9 \pm 0.17$ |
| Conv-LoRA | 4.02 / 0.63% | $\mathbf{84.7} \pm 0.69$ | $\mathbf{88.0} \pm 1.08$ | $\mathbf{75.3} \pm 0.89$ | $\mathbf{3.4} \pm 0.10$ | $\mathbf{87.1} \pm 1.70$ | $\mathbf{88.8} \pm 1.43$ | $\mathbf{71.6} \pm 3.87$ | $\mathbf{1.5} \pm 0.43$ | $93.7 \pm 0.18$ | $96.5 \pm 0.15$ | $87.2 \pm 0.12$ | $0.9 \pm 0.18$ |

Table 5: Quantitative results for Polyp Segmentation.

**Skin Lesion Segmentation**:

| Method | # Params (M) / Ratio(%) | ISIC 2017 | | | |
|---|---|---|---|---|---|
| | | Jac $\uparrow$ | T-Jac $\uparrow$ | Dice $\uparrow$ | Acc $\uparrow$ |
| *Transfuse (Zhang et al., 2021)* | 26.30 / 100% | 80.1 | 74.5 | 87.5 | 94.7 |
| decoder-only | 3.51 / 0.55% | 69.7 | 56.6 | 79.5 | 91.2 |
| BitFit | 3.96 / 0.62% | $76.4 \pm 0.45$ | $68.2 \pm 0.70$ | $84.7 \pm 0.35$ | $93.5 \pm 0.16$ |
| Adapter | 3.92 / 0.61% | $76.7 \pm 0.66$ | $68.0 \pm 1.09$ | $85.0 \pm 0.56$ | $93.6 \pm 0.17$ |
| VPT | 4.00 / 0.62% | $76.9 \pm 0.94$ | $68.4 \pm 1.44$ | $85.1 \pm 0.75$ | $93.7 \pm 0.43$ |
| LST | 11.49 / 1.77% | $76.4 \pm 1.05$ | $66.7 \pm 2.08$ | $84.9 \pm 0.79$ | $93.5 \pm 0.30$ |
| SAM-Adapter | 3.98 / 0.62% | $76.1 \pm 0.45$ | $67.2 \pm 0.77$ | $84.6 \pm 0.37$ | $93.4 \pm 0.18$ |
| SSF | 4.42 / 0.69% | $76.6 \pm 0.19$ | $68.3 \pm 0.62$ | $85.0 \pm 0.14$ | $93.6 \pm 0.12$ |
| LoRA | 4.00 / 0.62% | $76.6 \pm 0.23$ | $68.6 \pm 0.32$ | $84.9 \pm 0.22$ | $93.6 \pm 0.03$ |
| Conv-LoRA | 4.02 / 0.63% | $\mathbf{77.6} \pm 0.57$ | $\mathbf{69.6} \pm 0.90$ | $\mathbf{85.7} \pm 0.36$ | $\mathbf{93.9} \pm 0.18$ |

Table 6: Quantitative results for Skin Lesion Segmentation.

**Camouflaged Object Segmentation**:

| Method | # Params (M) / Ratio (%) | CHAMELEON | | | | CAMO | | | | COD10K | | | |
|---|---|---|---|---|---|---|---|---|---|---|---|---|---|
| | | $S_\alpha \uparrow$ | $E_\phi \uparrow$ | $F_\beta^\omega \uparrow$ | MAE $\downarrow$ | $S_\alpha \uparrow$ | $E_\phi \uparrow$ | $F_\beta^\omega \uparrow$ | MAE $\downarrow$ | $S_\alpha \uparrow$ | $E_\phi \uparrow$ | $F_\beta^\omega \uparrow$ | MAE $\downarrow$ |
| *SINet-v2 (Fan et al., 2022)* | 26.98 / 100% | 89.2 | 04.0 | 79.9 | 3.1 | 80.8 | 85.8 | 73.1 | 7.7 | 81.2 | 88.3 | 65.9 | 3.7 |
| decoder-only | 3.51 / 0.55% | 87.3 | 90.5 | 79.2 | 3.7 | 78.5 | 83.1 | 69.8 | 8.7 | 82.8 | 87.8 | 70.3 | 3.7 |
| BitFit | 3.96 / 0.62% | $93.2 \pm 0.3$ | $96.1 \pm 0.61$ | $88.7 \pm 0.72$ | $1.8 \pm 0.08$ | $86.8 \pm 0.33$ | $90.7 \pm 0.28$ | $81.5 \pm 0.19$ | $5.3 \pm 0.15$ | $91.1 \pm 0.11$ | $95.1 \pm 0.13$ | $85.5 \pm 0.10$ | $1.8 \pm 0.05$ |
| Adapter | 3.92 / 0.61% | $93.9 \pm 0.53$ | $96.6 \pm 0.55$ | $89.9 \pm 1.21$ | $1.7 \pm 0.18$ | $87.7 \pm 0.10$ | $91.3 \pm 0.40$ | $82.8 \pm 0.35$ | $5.0 \pm 0.18$ | $91.5 \pm 0.13$ | $95.3 \pm 0.13$ | $86.4 \pm 0.12$ | $1.7 \pm 0.01$ |
| VPT | 4.00 / 0.62% | $93.2 \pm 0.24$ | $96.1 \pm 0.42$ | $88.9 \pm 0.50$ | $1.9 \pm 0.04$ | $87.4 \pm 0.60$ | $91.4 \pm 0.68$ | $82.1 \pm 0.75$ | $5.0 \pm 0.25$ | $91.1 \pm 0.32$ | $94.9 \pm 0.23$ | $85.3 \pm 0.72$ | $1.8 \pm 0.09$ |
| LST | 11.49 / 1.77% | $92.0 \pm 0.19$ | $95.3 \pm 0.39$ | $87.2 \pm 0.58$ | $2.3 \pm 0.15$ | $83.3 \pm 0.28$ | $88.0 \pm 0.23$ | $77.1 \pm 0.02$ | $6.9 \pm 0.06$ | $88.4 \pm 0.16$ | $93.2 \pm 0.07$ | $80.8 \pm 0.26$ | $2.3 \pm 0.05$ |
| SAM-Adapter | 3.98 / 0.62% | $92.7 \pm 0.38$ | $95.9 \pm 0.42$ | $87.7 \pm 0.94$ | $2.0 \pm 0.19$ | $85.6 \pm 0.26$ | $89.6 \pm 0.55$ | $79.8 \pm 0.89$ | $5.9 \pm 0.16$ | $90.1 \pm 0.04$ | $94.1 \pm 0.17$ | $83.7 \pm 0.31$ | $2.0 \pm 0.03$ |
| SSF | 4.42 / 0.69% | $94.0 \pm 0.13$ | $\mathbf{97.0} \pm 0.11$ | $90.1 \pm 0.12$ | $\mathbf{1.5} \pm 0.11$ | $87.5 \pm 0.11$ | $91.4 \pm 0.16$ | $82.6 \pm 0.12$ | $5.0 \pm 0.11$ | $91.3 \pm 0.12$ | $95.1 \pm 0.08$ | $86.2 \pm 0.20$ | $1.7 \pm 0.04$ |
| LoRA | 4.00 / 0.62% | $93.8 \pm 0.17$ | $96.7 \pm 0.14$ | $89.8 \pm 0.20$ | $1.8 \pm 0.18$ | $88.0 \pm 0.24$ | $91.9 \pm 0.42$ | $82.8 \pm 0.16$ | $4.8 \pm 0.13$ | $91.5 \pm 0.10$ | $95.2 \pm 0.12$ | $86.4 \pm 0.16$ | $1.7 \pm 0.09$ |
| Conv-LoRA | 4.02 / 0.63% | $\mathbf{94.1} \pm 0.19$ | $96.9 \pm 0.15$ | $\mathbf{90.6} \pm 0.11$ | $1.6 \pm 0.10$ | $\mathbf{88.3} \pm 0.40$ | $\mathbf{92.4} \pm 0.31$ | $\mathbf{84.0} \pm 0.34$ | $\mathbf{4.5} \pm 0.19$ | $\mathbf{91.6} \pm 0.06$ | $\mathbf{95.5} \pm 0.08$ | $\mathbf{86.8} \pm 0.23$ | $\mathbf{1.6} \pm 0.02$ |

Table 7: Quantitative results for Camouflage Detection.

**Shadow Detection**:

| Method | # Params (M) / / Ratio(%) | SBU BER ↓ |
|---|---|---|
| *FDRNet(Zhu et al., 2021)* | 10.77 / 100% | 3.56 |
| decoder-only | 3.51 / 0.55% | 14.58 |
| BitFit | 3.96 / 0.62% | $3.16 \pm _{0.128}$ |
| Adapter | 3.92 / 0.61% | $2.84 \pm _{0.093}$ |
| VPT | 4.00 / 0.62% | $2.70 \pm _{0.055}$ |
| LST | 11.49 / 1.77% | $3.18 \pm _{0.012}$ |
| SAM-Adapter | 3.98 / 0.62% | $3.14 \pm _{0.063}$ |
| SSF | 4.42 / 0.69% | $3.19 \pm _{0.046}$ |
| LoRA | 4.00 / 0.62% | $2.74 \pm _{0.079}$ |
| Conv-LoRA | 4.02 / 0.63% | $\mathbf{2.54} \pm _{0.081}$ |

Table 8: Quantitative results for Shadow Detection.

**Leaf Segmentation**:

| Method | # Params (M) / Ratio (%) | Leaf | | |
|---|---|---|---|---|
| | | IoU ↑ | Dice ↑ | Acc ↑ |
| *DeepLabv3 (Chen et al., 2017b)* | 41.99 / 100% | 62.3 | 74.1 | 94.0 |
| decoder-only | 3.51 / 0.55% | 50.8 | 63.8 | 89.6 |
| BitFit | 3.96 / 0.62% | $71.4 \pm _{1.15}$ | $81.7 \pm _{1.01}$ | $95.5 \pm _{0.30}$ |
| Adapter | 3.92 / 0.61% | $72.1 \pm _{0.47}$ | $82.4 \pm _{0.36}$ | $95.3 \pm _{0.18}$ |
| VPT | 4.00 / 0.62% | $73.6 \pm _{0.26}$ | $83.8 \pm _{0.26}$ | $95.9 \pm _{0.14}$ |
| LST | 11.49 / 1.77% | $70.2 \pm _{0.87}$ | $81.1 \pm _{0.82}$ | $95.3 \pm _{0.35}$ |
| SAM-Adapter | 3.98 / 0.62% | $71.4 \pm _{0.20}$ | $82.1 \pm _{0.10}$ | $95.3 \pm _{0.05}$ |
| SSF | 4.42 / 0.69% | $71.5 \pm _{0.63}$ | $81.8 \pm _{0.44}$ | $95.5 \pm _{0.26}$ |
| LoRA | 4.00 / 0.62% | $73.7 \pm _{0.20}$ | $83.6 \pm _{0.13}$ | $95.7 \pm _{0.07}$ |
| Conv-LoRA | 4.02 / 0.63% | $\mathbf{74.5} \pm _{0.39}$ | $\mathbf{84.3} \pm _{0.34}$ | $\mathbf{96.0} \pm _{0.07}$ |

Table 9: Quantitative results for Leaf Segmentation.

**Road Segmentation**:

| Method | # Params (M) / Ratio (%) | Road | | |
|---|---|---|---|---|
| | | IoU ↑ | Dice ↑ | Acc ↑ |
| *LinkNet34MTL(Batra et al., 2019)* | 22.00 / 100% | 59.1 | 73.0 | 97.7 |
| decoder-only | 3.51 / 0.55% | 48.6 | 65.1 | 96.4 |
| BitFit | 3.96 / 0.62% | $60.6 \pm _{0.15}$ | $75.2 \pm _{0.11}$ | $97.5 \pm _{0.02}$ |
| Adapter | 3.92 / 0.61% | $61.5 \pm _{0.11}$ | $75.9 \pm _{0.12}$ | $97.6 \pm _{0.01}$ |
| VPT | 4.00 / 0.62% | $60.2 \pm _{1.87}$ | $74.9 \pm _{1.50}$ | $97.4 \pm _{0.22}$ |
| LST | 11.49 / 1.77% | $60.2 \pm _{0.26}$ | $74.9 \pm _{0.22}$ | $97.5 \pm _{0.01}$ |
| SAM-Adapter | 3.98 / 0.62% | $60.6 \pm _{0.06}$ | $75.2 \pm _{0.04}$ | $97.5 \pm _{0.01}$ |
| SSF | 4.42 / 0.69% | $61.6 \pm _{0.03}$ | $76.0 \pm _{0.02}$ | $97.6 \pm _{0.01}$ |
| LoRA | 4.00 / 0.62% | $62.2 \pm _{0.21}$ | $76.5 \pm _{0.18}$ | $97.6 \pm _{0.02}$ |
| Conv-LoRA | 4.02 / 0.63% | $\mathbf{62.6} \pm _{0.36}$ | $\mathbf{76.8} \pm _{0.27}$ | $\mathbf{97.7} \pm _{0.05}$ |

Table 10: Quantitative results for Road Segmentation.

# E  ADDITIONAL ABLATION

**The Rank of LoRA.** The performance of SAM for twelve-class segmentation could be improved by introducing more extra parameters (table 11). However, this may disobey the rule of 'parameter efficiency' of PEFT. Exploring the design of a more efficient way for introducing 'classification prior' for SAM is a worthwhile endeavor for the future.

| LoRA | #Params (M) | Test | |
|---|---|---|---|
| | | Acc ↑ | mIoU ↑ |
| $r = 3$ | 4.00 | 94.80 | 66.01 |
| $r = 6$ | 4.49 | 95.15 | 66.24 |
| $r = 12$ | 5.48 | 95.23 | 66.69 |
| $r = 24$ | 7.44 | **95.14** | **67.02** |

Table 11: The performance trend when increasing the rank $r$ of LoRA for twelve-class segmentation.

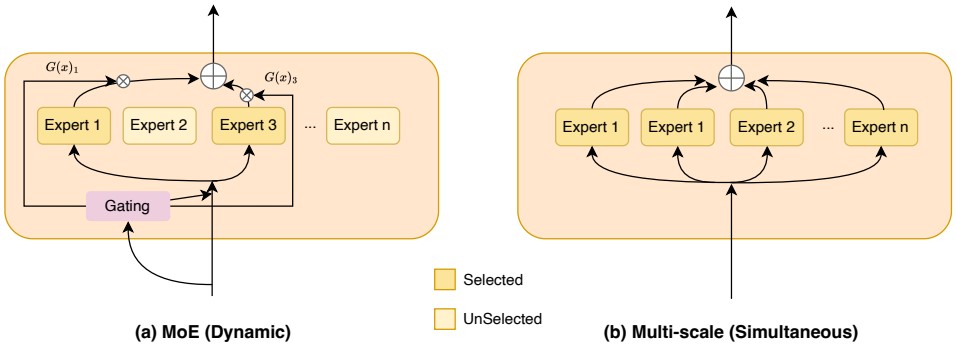

Figure 6: Comparison between (a) MoE, which uses the gating network to dynamically select the experts, and (b) Multi-scale, which selects all the experts simultaneously.

**Combined with Other PEFT Methods.** We also evaluate the combination of Conv-LoRA and VPT on Poly Segmentation in table 12. This combination achieves superior or competitive performance, demonstrating the potential of introducing Conv-LoRA in addition to other PEFT methods. As a future work, such a combination might motivate techniques that further reduce the number of trainable parameters while ensuring the enhanced performance.

| Method | # Params / Ratio(%) | CVC-ColonDB | | | | ETIS | | | |
|---|---|---|---|---|---|---|---|---|---|
| | | $S_\alpha \uparrow$ | $E_\phi \uparrow$ | $F_\beta^\omega \uparrow$ | MAE ↓ | $S_\alpha \uparrow$ | $E_\phi \uparrow$ | $F_\beta^\omega \uparrow$ | MAE ↓ |
| VPT | 4.00 / 0.62% | 85.0 | 88.4 | 75.0 | 3.6 | 86.5 | 87.4 | 70.1 | 1.9 |
| Conv-LoRA | 4.03 / 0.63% | 85.1 | 88.8 | 75.7 | **3.4** | 86.8 | **88.5** | 70.2 | **1.7** |
| Conv-LoRA+VPT | 4.23 / 0.66 % | **86.0** | **89.0** | **76.8** | 3.5 | **87.7** | 88.1 | **70.3** | **1.7** |

Table 12: Performance of combining Conv-LoRA and VPT on Polyp Segmentation.

**MoE-Conv vs. Blocks with Various Kernel sizes.** We further compare the design of our MoE-Conv and convolutional blocks with various kernel sizes. Convolutional blocks with various kernel sizes are commonly used in multi-scale feature fusion. Here we follow the Inception module proposed by (Szegedy et al., 2015), which consists of multiple kernel sizes with $1 \times 1$, $3 \times 3$, $5 \times 5$. To ensure a fair relative comparison, we opt for four experts in MoE-Conv to align with the branch number of the Inception structure. We conduct experiments on Leaf Dataset and ISIC 2017 Dataset twice for evaluation.

| Method | # Params / Ratio(%) | Leaf | | | ISIC 2017 | | | |
|---|---|---|---|---|---|---|---|---|
| | | IoU ↑ | Dice ↑ | Acc ↑ | Jac ↑ | T-Jac ↑ | Dice ↑ | Acc ↑ |
| Inception | 4.01 / 0.63% | 72.1 | 82.3 | 95.0 | 76.9 | 68.5 | 84.9 | 93.3 |
| MoE-Conv | 4.01 / 0.63% | **74.0** | **83.8** | **95.6** | **77.4** | **69.5** | **85.5** | **93.9** |

Table 13: MoE-Conv vs. Blocks with Various Kernel sizes (indicated as 'Inception').

In table 13, MoE-Conv outperforms the design of convolutional blocks with various kernel sizes. We attribute this to **convolutions with various kernel sizes only operate on the default feature scale**. Maybe a larger kernel size can inject some approximate local prior in features of a smaller scale, but it probably can't bring local prior into features of larger scales than the default. The features in ViT are downscaled 16x from the original image, so injecting priors in larger scale features are generally more useful, as evidenced in table 4, where the optimal scale is usually larger than the default scale.

**Computational Cost.** In table 14, we compare the training / inference speed and per epoch training time with the ISIC 2017 dataset and a single V100 GPU. While Conv-LoRA is slower in speed than others, it achieves robust performance gains across various semantic segmentation tasks (table 1). We find that the main cost comes from the upscale and downscale operations. A possible future direction is exploring how to inject local prior without explicitly scaling up and down features. We also notice that there are some other orthogonal works, like token merging (Liang et al., 2022; Bolya et al., 2022), that may accelerate Conv-LoRA.

| Method | Training | | Inference |
|---|---|---|---|
| | iter/s | min/epoch | iter/s |
| BitFit | 1.82 | 18.3 | 4.62 |
| Adapter | 1.73 | 19.3 | 4.38 |
| VPT | 1.41 | 23.6 | 4.39 |
| LoRA | 1.64 | 20.3 | 4.62 |
| Conv-LoRA | 1.22 | 27.3 | 3.64 |

Table 14: Computational cost comparison.

## F    LOCAL PRIOR ANALYSIS

We elaborate on 'SAM's local prior assumption' mentioned in our abstract in section 4.3. Additionally, as our Conv-LoRA is designed to reinforce existing inductive biases within the features, we investigate the presence of these biases in the features.

We use two metrics: 1) Mean attention distance, which reflects the extent of **local or global information** that a self-attention layer is aggregating (Dosovitskiy et al., 2020; Raghu et al., 2021). We calculate the attention distance for each attention head, defined as the average distance between the position of the query patch and the locations to which it attends, weighted by the attention weights. Then we calculate the average attention distance for each layer by computing the mean across 500 randomly sampled images from downstream semantic segmentation tasks. 2) Relative log amplitudes (Park & Kim, 2022), which reflects whether the model tends to reduce or amplify **high-frequency signals (e.g., edges, textures)** in feature map. We compute the relative log amplitudes of the Fourier-transformed feature map in each layer and average them across layers. Then we calculate the mean of the relative log amplitudes over 100 randomly sampled segmentation images.

**Inductive biases of the features.** Extensive pre-training on large-scale datasets has been demonstrated to equip ViTs with inductive biases (Dosovitskiy et al., 2020; Raghu et al., 2021). SAM's large-scale segmentation pretraining further amplifies these inductive biases within the features.

To evaluate the inductive biases of the features, we conduct comprehensive comparisons among randomly initialized ViT, MAE pretrained ViT, and SAM ViT. We use mean attention distance and relative log amplitudes for evaluation. In fig. 7, as the training progresses from randomly initialized ViT to MAE ViT and ultimately to SAM ViT, we observe a consistent trend: the mean attention distance decreases, while the high-frequency signals in the features increase, indicating a focus on local information. **These findings further confirm the presence of inductive biases in features following large-scale pre-training. Furthermore, the findings affirm the rationality of Conv-LoRA in reinforcing the inherent inductive biases within the features.**

## G    MOE ANALYSIS

**What does MoE learn?** MoE learns to **dynamically select an appropriate scale for injecting local priors based on input features.** To illuminate its functionality, we conduct an analysis by

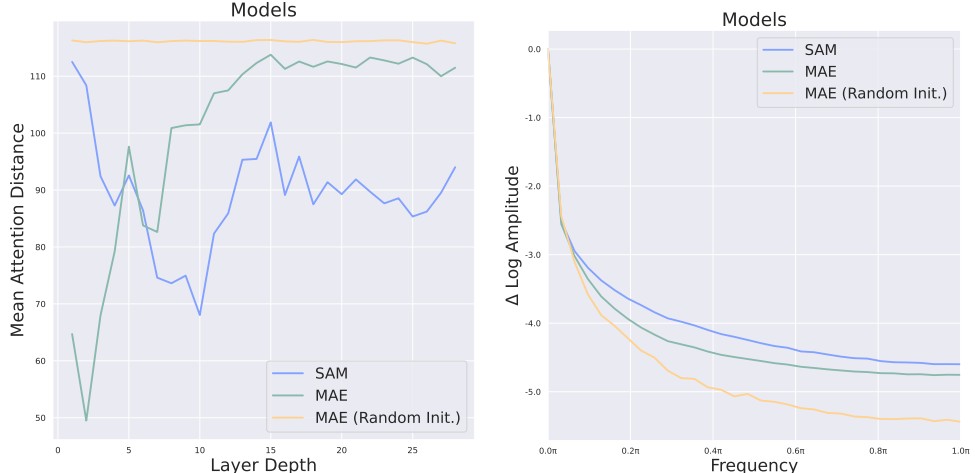

Figure 7: Evaluation of inductive biases within features. *Left:* Mean attention distance average over all attention heads. *Right:* Relative log amplitudes of Fourier transformed feature maps. Compared to randomly initialized ViT, SAM and MAE could learn the inductive biases from large-scale data pre-training. And SAM could further amplifies these biases within the features.

tracking the frequency of each expert's selection across different datasets during inference. The detailed results are presented in fig. 8.

Notably, distinct datasets exhibit preferences for different experts. For instance, in Leaf Segmentation, MoE tends to favor experts with upsampling ratios of 3 and 4, while on the ISIC 2017 dataset, it tends to select the expert with an upsampling ratio of 2. Connecting these insights with the optimal scale ablation results in table 4, it become evident that MoE selects the expert adaptively for each sample, and consequently, the distributions of MoE selection across datasets reflect their different data distributions. This observation supports that MoE behaves adaptively and effectively with respect to each sample's property. **This adaptability reinforces the significance of MoE in tailoring its selection to the unique characteristics of diverse datasets, enhancing its effectiveness in local prior injection.**

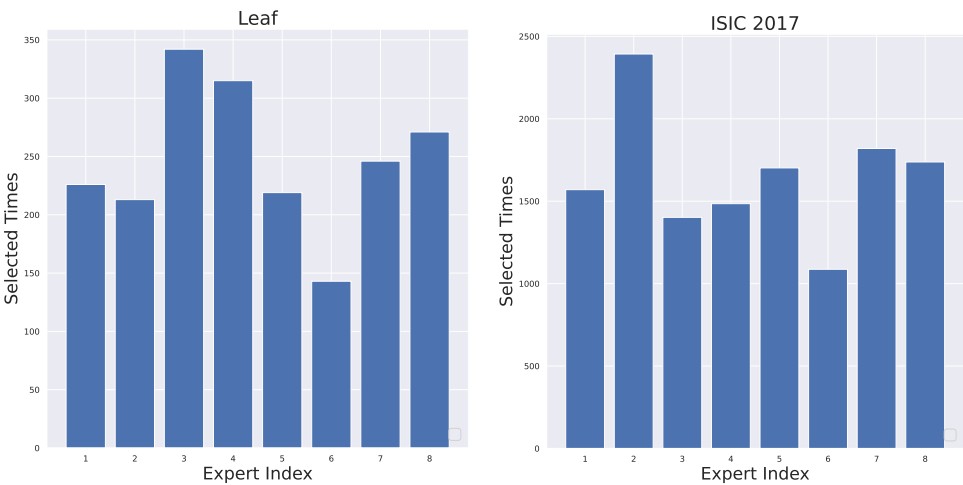

Figure 8: The number of times each expert is selected during inference for on Leaf Dataset and ISIC 2017 Dataset. Distinct datasets exhibit preferences for different experts.

## H LOW-DATA REGIME EXPERIMENTS

Convolutional layers could introduce inductive biases, thereby enhancing data efficiency during fine-tuning. Therefore, we undertake experiments in a low-data regime to validate whether Conv-LoRA can indeed improve data efficiency in semantic segmentation.

Specifically, we conduct experiments under the few-shot setting, wherein acquiring data for downstream tasks is challenging, and only a limited number of training samples per task are available. Our experiments are performed on the Trans10K-v2 dataset, which is used for twelve-class transparent object segmentation. We randomly select the training samples, to meet the settings of 1, 2, 4, 8, and 16 shots (i.e., the number of labeled training examples per class), and the experiments are run for 100 epochs.

| Method | # Params / Ratio(%) | Shot | mIoU ↑ |
|--------|---------------------|------|--------|
| LoRA | 4.00 / 0.62% | 1/2/4/8/16 | 13.32/18.01/22.70/25.89/38.11 |
| Conv-LoRA | 4.02 / 0.63% | 1/2/4/8/16 | **14.53/19.81/23.05/26.02/38.63** |

Table 15: Results of few-shot learning on multi-class semantic segmentation. 'Shot' indicates the number of labeled training examples per class.

In table 15, the performance improvement of Conv-LoRA compared to LoRA, is particularly pronounced in an extremely low-data setting (e.g., 1-shot). The result demonstrates that **the introduction of inductive biases in Conv-LoRA contribute to improved data efficiency**.

## I MORE VISUALIZATION RESULTS

**Conv-LoRA vs. LoRA Feature.** We visualize the feature maps from the fine-tuned SAM's image encoder, which incorporates LoRA and our Conv-LoRA respectively. In fig. 9, the image features from SAM's image encoder equipped with Conv-LoRA could provide more fine-grained information, e.g., slim edges, which is beneficial to later mask prediction. This further demonstrates the effectiveness of reinforcing the image-related local priors with network architecture.

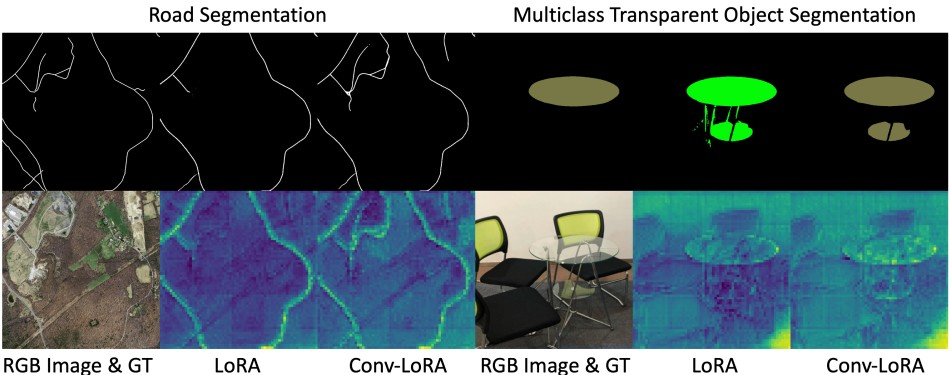

Figure 9: Feature map visualization for binary-class and multi-class semantic segmentation. Compared to LoRA, applying Conv-LoRA to SAM's image encoder could capture more fine-grained details.

**Mask prediction.** We also provide more visualization results for the mask prediction across various datasets when applying different PEFT methods (VPT, LoRA and Conv-LoRA). This further confirms the superiority of our Conv-LoRA.

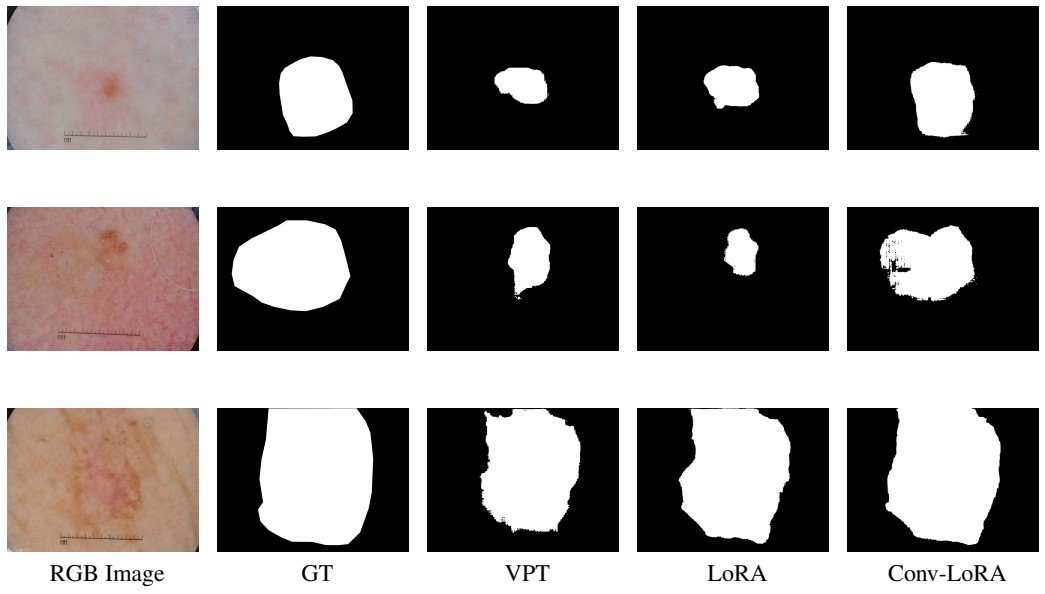

Figure 10: Visualization Results on Skin Lesion Segmentation.

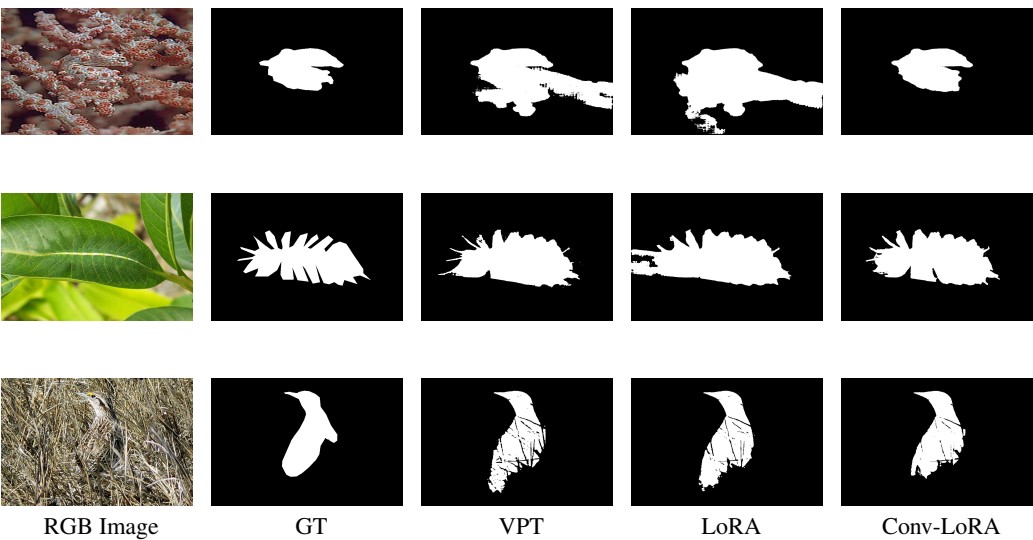

Figure 11: Visualization Results on Camouflaged Object Segmentation.

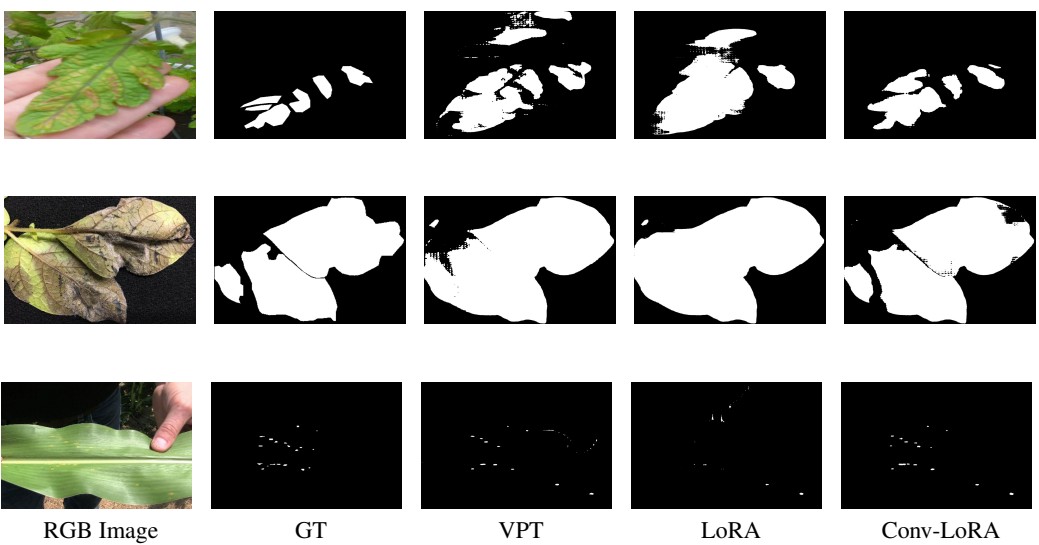

Figure 12: Visualization Results on Leaf Disease Segmentation.

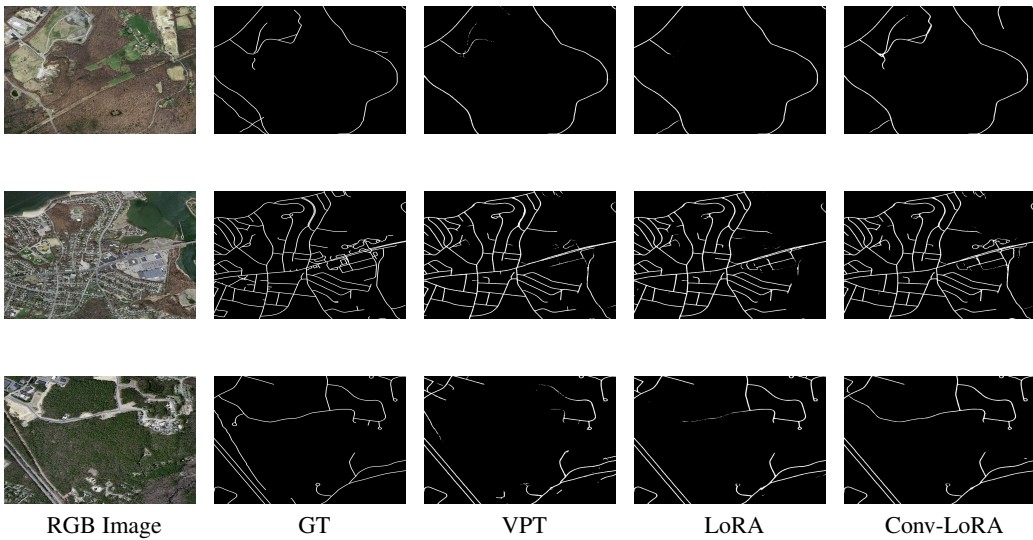

Figure 13: Visualization Results on Remote Sensing.

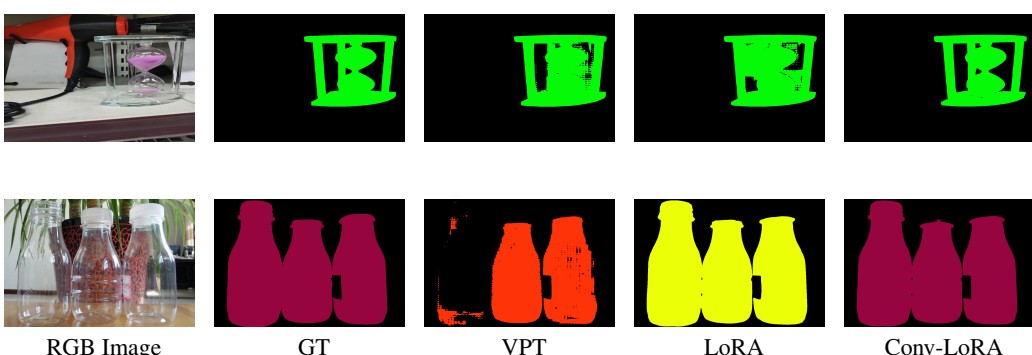

Figure 14: Visualization Results on Multi-class Transparent Object Segmentation.

