# OpenReview forum: "Convolution Meets LoRA: Parameter Efficient Finetuning for Segment Anything Model"
_ICLR.cc/2024/Conference — ICLR 2024 poster_

### Official Review · Reviewer_2Lzf · 2023-11-04

**Soundness:** 3 good
**Presentation:** 3 good
**Contribution:** 2 fair
**Rating:** 6
**Confidence:** 3

**Summary:**

This paper is an improved version of LoRA (low-rank adaptation) that aims to adapt pre-trained Segment Anything Models (SAM) across a diverse array of semantic segmentation tasks. The proposed Conv-LoRA adds an extra convolution operation between the encoder and decoder. Freezing the pre-trained SAM, the encoder, decoder, and extra convolution are trainable to adapt SAM to downstream semantic segmentation tasks. The trainable part of this framework is lightweight. Compared with LoRA and other parameter-efficient fine-tuning approaches, the proposed Conv-LoRA shows better performance for semantic segmentation tasks in medical natural imaging, agriculture, and remote sensing.

**Strengths:**

+ The manuscript is composed with clarity, presenting a concept that is both coherent and well-motivated.
+ The experimental setting is clearly stated, and the authors have conducted comparisons with not only LoRA but also a wide array of baseline methods.
+ The scope of experimentation is thorough. The proposed Conv-LoRA notably surpasses LoRA and a range of other methods aiming for efficient fine-tuning. It also appears to outperform full fine-tuning 100% parameters.
+ It appears that incorporating convolution operations can improve the SAM features in terms of fine-grained information, such as slim edges, and semantic information (this might be due to fine-tuning).

**Weaknesses:**

- Lack of comparison with training segmentation model from scratch (random initialization). It remains unclear whether the pre-trained SAM model helps transfer to downstream semantic segmentation tasks.

**Questions:**

1. The Mixture of Experts in introduced in the paper seems to be multi-scale strategies that are commonly used in image segmentation. It is unclear whether the added complexity of this method is warranted. A more straightforward explanation would likely render the technique more impactful and accecible.

---

> ### Author Response · Authors · 2023-11-22
> **Replies to Reviewer 2Lzf**
>
> Thank you for your detailed review on our paper. We will address each of your comments thoroughly in the following sections.
>
> **Q1: Comparison with training segmentation model from scratch.**
>
> We conducted a performance comparison (**Table 14 in Appendix F**) between SAM training from scratch and Conv-LoRA on two datasets: ISIC 2017 Dataset and Leaf Segmentation Dataset. The substantial performance gaps observed underscore the considerable assistance provided by **SAM's pretraining knowledge in enhancing downstream task performance**. Moreover, the pretraining-finetuning paradigm has been proved useful in prior studies [1-3].
>
> *ISIC2017 Dataset*
>
> | Method                     | #Params (M) | Jac      | T-Jac    | Dice     | Acc      |
> | -------------------------- | ----------- | -------- | -------- | -------- | -------- |
> | SAM (trained from scratch) | 641.09      | 73.8     | 63.8     | 82.5     | 92.4     |
> | Conv-LoRA                  | 4.02        | **77.6** | **69.6** | **85.7** | **93.9** |
>
> *Leaf Segmentation Dataset*
>
> | Method                     | #Params (M) | IoU       | Dice     | Acc      |
> | -------------------------- | ----------- | --------- | -------- | -------- |
> | SAM (trained from scratch) | 641.09      | 52.1      | 65.5     | 89.8     |
> | Conv-LoRA                  | 4.02        | **74.5** | **84.3** | **96.0** |
>
> ------
>
> **Q2: MoE vs. common multi-scale strategies.**
>
> A straightforward multi-scale strategy involves using multiple convolution branches and subsequently aggregating their features. In comparison, MoE proves to be more efficient due to its utilization of conditional computation, selectively activating only one branch during each forward pass. This conclusion is supported by the ablation study presented in **Table 3**, where MoE demonstrates faster training speed and reduced memory consumption. We also clarify the motivation of introducing MoE in **Section 3.1**.
>
> An alternative multi-scale strategy entails integrating additional convolution networks alongside SAM to extract multi-scale features, akin to the ViT-adapter design [4]. However, this approach is no longer parameter-efficient, as it requires full fine-tuning the pre-trained backbone, which introduces significantly more parameters than LoRA, leading to increased training complexity. In contrast to these explicit multi-scale strategies, **Conv-LoRA operates more implicitly within ViT, which inherently processes features predominantly at a single scale.** This implicit integration distinguishes Conv-LoRA as a parameter-efficient alternative in addressing multi-scale considerations.
>
> ------
>
> References:
>
> [1] Dosovitskiy A, et al. "An image is worth 16x16 words: Transformers for image recognition at scale." arXiv preprint arXiv:2010.11929, 2020.
>
> [2] Kaiming He, et al. "Masked autoencoders are scalable vision learners." Proceedings of the IEEE/CVF conference on computer vision and pattern recognition, pp. 16000–16009, 2022.
>
> [3] Liu Z, et al. "Swin transformer: Hierarchical vision transformer using shifted windows." Proceedings of the IEEE/CVF international conference on computer vision. 2021: 10012-10022.
>
> [4] Chen Z, et al. "Vision transformer adapter for dense predictions." arXiv preprint arXiv:2205.08534, 2022.

---

> > ### Comment · Reviewer_2Lzf · 2023-11-22
> >
> > Thank you for the well-prepared response and revision, which have addressed my earlier concerns. A following-up suggestion is to include `learning from scratch` performance in the main paper, perhaps adding one more row in the table instead of hiding it in the appendix. It is because the scratch result is an important reference for a transfer learning study.

---

> > > ### Author Response · Authors · 2023-11-23
> > > **Thank you for your valuable suggestion!**
> > >
> > > Thank you for your valuable suggestion! Due to the deadline for discussion approaching, we may not have sufficient time to complete all the experiments of `learning from scratch` on the datasets shown in our main table. Nevertheless, we will add the results of `learning from scratch` into the main table in the camera-ready version.
> > >
> > > And we are also delighted that our response have addressed your earlier concerns.

---

### Official Review · Reviewer_LN6g · 2023-11-04

**Soundness:** 3 good
**Presentation:** 3 good
**Contribution:** 3 good
**Rating:** 6
**Confidence:** 4

**Summary:**

This paper proposes an improvement to the Segment Anything Model (SAM) by incorporating fittings to alleviate some of the original model's deficiencies, together with algorithmic novelties to improve performance. Improvements to SAM are claimed in specialized domains (e.g medical images) and in improving fine grained semantic predictions. Methods are proposed to build upon LoRA (Low Rank Approximation) from NLP, which is modified to incorporate a Mixture of Experts setup with convolutional processing (what they call conv-LoRA). This way, it is claimed that the model is able to aggregate signals from multiple experts that can learn image priors in a specialized way, all while learning with minimal computational overhead, or what they call parameter efficient fine tuning. A few other modifications are used to boost performance and usability (freezing prompts, allowing for a classification head for multiclass classification) over SAM.

Results show improvements over other parameter efficient fine tuning models (e.g. VPT), with little overhead from the model's conv fittings.

**Strengths:**

+ Solidly written paper, well motivated ideas for SAM improvement
+ Novelty in the use of MoE
+ Improvements in producing fine grained classification (e.g. edges), and in specialized domains

**Weaknesses:**

- Purely empirical work. No reasoning is given as to how the model improves performance.
- MoE is not well motivated. I looked up the original paper for insight, which clarifies things somewhat, but the paper in question should describe it better.
- Performance improvement is very marginal.

**Questions:**

Ablations: I am curious about the effect of various model components on performance.
- MoE looks a bit opaque. What priors is it learning, and how do we grasp what it is doing?
- Can the authors give more reasoning for freezing the prompt encoder? I quite liked the prompt encoder in the original model.

---

> ### Author Response · Authors · 2023-11-22
> **Replies to Reviewer LN6g**
>
> Thank you for your thoughtful review on our paper. We will comprehensively respond to each of your comments in the following sections.
>
> **Q1: The reasons of performance improvements.**
>
> Our enhancements are rooted in two pivotal aspects. First, SAM relies on the plain ViT architecture, which is known for its absence of image-related local priors. **Conv-LoRA tackles this limitation by introducing convolution operations that incorporate hard-coded local priors.** The integration of local priors proves beneficial for segmentation tasks demanding pixel-level predictions.
>
> Second, SAM's segmentation pretraining causes the ViT to emphasize low-level features while neglecting significant high-level semantic information essential for semantic segmentation. In contrast, **Conv-LoRA, through finetuning, facilitates the ViT in reclaiming the capacity to capture nuanced high-level semantic information.**
>
> ------
>
> **Q2: Motivation of MoE.**
>
> We leverage MoE to **efficiently** **infuse local priors into ViT**. As highlighted in **Table 4** of our paper, the optimal feature scale for injecting local priors varies across different datasets, necessitating a multi-scale strategy. While a conventional approach involves employing multiple branches to simultaneously inject local priors at different scales and aggregating them, this method incurs higher computational costs compared to MoE. The efficiency of MoE lies in its ability to selectively activate sparse experts, minimizing computational overhead. Given our emphasis on efficient finetuning, we opt for MoE as a judicious choice for injecting local priors into ViT. We also clarify the motivation of introducing MoE in **Section 3.1**.
>
> ------
>
> **Q3: Marginal improvements.**
>
> Our approach showcases notable improvements on some datasets. For instance, on the ETIS dataset, as illustrated in **Table 5**, Conv-LoRA exhibits a performance boost of approximately 2% across three metrics when compared to LoRA. In certain cases, even a 0.2% improvement, as observed on the SBU dataset in **Table 1**, is deemed significant within the context of semantic segmentation literatures [1] [2].
>
> It is essential to emphasize that **our method adheres to a universal setting, validated across a diverse array of semantic segmentation tasks**. This universality underscores the versatility and efficacy of Conv-LoRA, making it a robust choice for various applications in the realm of semantic segmentation.
>
> ------
>
> **Q4: What priors is MoE learning and how we grasp what it is doing?**
>
> **MoE learns to dynamically select an appropriate scale for injecting local priors based on input features.** To illuminate its functionality, we conducted an analysis by tracking the frequency of each expert's selection across different datasets during inference. The detailed results are presented in **Figure 14 in Appendix H**. Notably, **distinct datasets exhibit preferences for different experts.** For instance, in Leaf Segmentation, MoE tends to favor experts with upsampling ratios of 3 and 4, while on the ISIC 2017 dataset, it tends to select the expert with an upsampling ratio of 2. Connecting these insights with the optimal scale ablation results in **Table 4**, it become evident that MoE selects the expert adaptively for each sample, and consequently, the distributions of MoE selection across datasets reflect their different data distributions. This observation supports that MoE behaves adaptively and effectively with respect to each sample’s property. **This adaptability reinforces the significance of MoE in tailoring its selection to the unique characteristics of diverse datasets,** **enhancing its effectiveness in local prior injection.**
>
> ------
>
> **Q5: Reason of freezing the prompt encoder.**
>
> While the prompt encoder in SAM provides opportunities for seamless integration with other modules, such as object detectors, within larger systems, **certain applications may necessitate a standalone end-to-end model for deployment**. In this work, our focus is specifically on these application scenarios, **aiming for a model that operates independently without reliance on external prompts**. For simplicity, we have opted to remove the prompt encoder during both training and inference stages. The exploration of joint finetuning with the prompt encoder is left for future studies.
>
> ------
>
> References:
>
> [1] Zheng Q, et al. "Distraction-aware shadow detection" .Proceedings of the IEEE/CVF Conference on Computer Vision and Pattern Recognition. 2019: 5167-5176.
>
> [2] Zhu L, et al. "Mitigating intensity bias in shadow detection via feature decomposition and reweighting". Proceedings of the IEEE/CVF International Conference on Computer Vision. 2021: 4702-4711.

---

> > ### Comment · Reviewer_LN6g · 2023-11-22
> >
> > Thanks for the clarifications. The response to reviewer concerns is interesting and fairly convincing, especially the additions of mean attention distance and performance in low data regime.

---

> > > ### Author Response · Authors · 2023-11-23
> > > **Thank you for your feedback!**
> > >
> > > Thank you for your feedback! We are delighted that our response have addressed your concerns, and we appreciate your acknowledgment of the experiments we added.

---

### Official Review · Reviewer_3rwQ · 2023-11-06

**Soundness:** 3 good
**Presentation:** 3 good
**Contribution:** 3 good
**Rating:** 6
**Confidence:** 4

**Summary:**

The authors proposed a parameter-efficient fine-tuning approach, i.e., a Conv-LoRA module combining trainable convolutional parameters with MOE scheme. It is developed to overcome the SAM’s performance drop when applied to specialized domains such as medical imagery and remote sensing. The Conv-LoRA module is integrated into the plain ViT encoder, enhancing SAM’s local prior assumption and its ability to learn high-level image semantics. Several previous parameter-efficient fine-tuning approaches are included in the comparison study. The proposed method enables efficient adaptation (with superior results) to real-world semantic segmentation tasks across various benchmarks and domains.

**Strengths:**

+ a novel LoRA-like add-on module for efficient parameter tuning for SAM, a typical large-vision model.
+ Superior results of the proposed methods are reported in comparison to vanilla LoRA, VPT, and other adaptor-based methods.

**Weaknesses:**

- The motivation for introducing the combination of MOE with convolutional parameters as a couple is not clear to me. It is not clear how each of them will benefit the performance as an add-on to the vanilla LoRA.
- The proposed method reminds me of the inception structure from the GoogLeNet. What will be the difference between the design of multiple down-scale+conv+up-scale blocks and convolutional blocks with various kernel sizes? Is MOE really necessary here? Will simple addition(or average) work?
- It will be helpful to clarify how different the training procedure with the add-on module will be in comparison to the original SAM training process.
- It will also be helpful to have a comparison in computational cost for those parameter-efficient tuning methods.
- As shown in Table 4, the scales vary amongst different datasets. Will this require extra tuning efforts for picking a suitable scale (experts)? Again, will the way how different sizes are combined in the inception structure be a better option?

**Questions:**

See weaknesses

---

> ### Author Response · Authors · 2023-11-22
> **Replies to Reviewer 3rwQ: first part**
>
> We appreciate your detailed and valuable review on our paper. We will address each of your comments thoroughly in the following sections.
>
> **Q1: Motivation of MoE + convolution as a couple.**
>
> Convolution is to inject the local prior into the plain ViT, and MoE is to help to do so efficiently. **Table 4** in our paper reveals that different datasets exhibit varying optimal scales. While a multi-scale strategy is a straightforward approach, our results indicate that employing MoE proves to be a more efficient solution. As demonstrated in **Table 3**, the utilization of MoE demonstrates notable advantages in terms of training speed (0.71 vs. 0.46 iterations per second) and lower training memory consumption (21.7GB vs. 23.4GB) compared to the multi-scale strategy. We also clarify the motivation of introducing MoE in **Section 3.1**.
>
> -------------------------------------------------------------------------------------------------------------------------------------------------------------
>
> **Q2 (1): What will be the difference between multiple down-scale+conv+up-scale blocks vs. convolutional blocks with various kernel sizes?**
>
> **The key difference lies in how they operate on feature scales**. Convolutions with different kernel sizes primarily function within the default feature scale. While a larger kernel size may approximate the injection of local priors into features of a smaller scale, **it may not effectively introduce local priors into features of larger scales than the default**. Given that the features in the Vision Transformer (ViT) are downscaled by a factor of 16 from the original image, the injection of priors into larger scale features proves to be generally more beneficial, as substantiated by the findings presented in **Table 4**, where the optimal scale is usually larger than the default scale.
>
> Moreover, we conducted a performance comparison between various kernel sizes, following the Inception structure [1], and our proposed design. To ensure a fair evaluation, our design incorporates four experts, aligning with the branch number of the Inception structure. Experimental results (**Table 13 of Appendix F**) on ISIC 2017 Dataset and Leaf Dataset, each performed twice and averaged, unequivocally demonstrate the superior performance of our design over the Inception structure.
>
> *ISIC 2017 Dataset*:
>
> | Method    | #Params (M) | Jac      | T-Jac    | Dice     | Acc      |
> | --------- | :---------: | -------- | -------- | -------- | -------- |
> | Inception |    4.01     | 76.9     | 68.5     | 84.9     | 93.3     |
> | Conv-LoRA |    4.01     | **77.4** | **69.5** | **85.5** | **93.9** |
>
> *Leaf Dataset:*
>
> | Method    | #Params (M) | IoU      | Dice     | Acc      |
> | --------- | :---------: | -------- | -------- | -------- |
> | Inception |    4.01     | 72.1     | 82.3     | 95.0     |
> | Conv-LoRA |    4.01     | **74.0** | **83.8** | **95.6** |
>
> -------------------------------------------------------------------------------------------------------------------------------------------------------------
>
> **Q2 (2): Is MoE necessary? Will simple addition (or average work)?**
>
> In contrast to the approach of employing multiple branches followed by addition or averaging, where all branches are activated for every sample, the use of sparsely gated MoE proves to be a more efficient method without incurring a proportional increase in computational costs. As highlighted in **Table 3** in our paper, **MoE demonstrates superior efficiency in terms of both training speed and memory consumption.** This efficiency holds significant importance for Conv-LoRA, particularly as our focus lies in the efficient fine-tuning of SAM. Additionally, **MoE outperforms simple multi-scale addition as shown in Table 3.**
>
> -------------------------------------------------------------------------------------------------------------------------------------------------------------
>
> **Q3: Compare the training procedure with original SAM.**
>
> SAM’s image encoder was initialized from MAE pretrained ViT, and SAM was pretrained on billion masks and 11 million images. The training is a promptable segmentation task with a set of geometric prompts, including mask, point, and box. Given one image and a prompt, SAM predicts three nested masks (for the whole, part, and subpart), computes a linear combination of focal loss and dice loss for each mask, and backpropagates solely the minimal loss among the three predictions.
>
> We aim to do parameter efficient finetuning (PEFT) of pretrained SAM on downstream tasks. During training, all the parameters in SAM’s image encoder are frozen except for the small amount of Conv-LoRA parameters. Additionally, we exclude the prompt encoder to transform SAM into an end-to-end model, eliminating the reliance on external prompts. In this finetuning process, the model directly predicts semantic segmentation masks from a given image, supervised by the ground truth masks specific to the downstream tasks.

---

> > ### Comment · Reviewer_3rwQ · 2023-12-04
> > **rebuttal feedback**
> >
> > I thank the authors for the detailed response. It clarifies most of my concerns, and I will keep my original rating, leaning toward to acceptance.

---

> ### Author Response · Authors · 2023-11-22
> **Replies to Reviewer 3rwQ: second part**
>
> **Q4: Computational cost comparison.**
>
> The comparison below evaluates the training and inference speed, along with the per-epoch training time, utilizing the ISIC2017 dataset on a single V100 GPU (**Table 15 of Appendix F**). Despite Conv-LoRA exhibiting a slower processing speed compared to other methods, **it demonstrates consistent performance improvements across diverse semantic segmentation tasks**, as illustrated in **Table 1** of our paper. Our analysis indicates that the primary computational cost arises from the upscale and downscale operations. A potential future direction is investigating methods to inject local priors without the explicit scaling up and down of features. Additionally, we notice the presence of orthogonal works, such as token merging [2] [3], which may offer ways for accelerating Conv-LoRA.
>
> *Training*:
>
> | Method    | iter/s | min/epoch |
> | --------- | ------ | :-------: |
> | BitFit    | 1.82   |   18.3    |
> | Adapter   | 1.73   |   19.3    |
> | VPT       | 1.41   |   23.6    |
> | LoRA      | 1.64   |   20.3    |
> | Conv-LoRA | 1.22   |   27.3    |
>
> *Inference*:
>
> | Method    | iter/s |
> | --------- | ------ |
> | BitFit    | 4.62   |
> | Adapter   | 4.38   |
> | VPT       | 4.39   |
> | LoRA      | 4.62   |
> | Conv-LoRA | 3.64   |
>
> -------------------------------------------------------------------------------------------------------------------------------------------------------------
>
> **Q5: Extra tuning efforts for picking suitable experts? Again, will the way how different sizes are combined in the inception structure be a better option?**
>
> We employ MoE to **automate the selection of appropriate scales**, obviating the need for additional tuning efforts. In our experiments, we utilize 8 experts that cover a range of 8 scales, a configuration we have found to be effective for many datasets. As previously discussed, **our rationale for favoring MoE over the inception structure stems from its ability to handle scales larger than the default**, a capability that the inception structure lacks.
>
> -------------------------------------------------------------------------------------------------------------------------------------------------------------
>
> References:
>
> [1] Szegedy C, et al. “Going deeper with convolutions”, Proceedings of the IEEE conference on computer vision and pattern recognition. 2015: 1-9.
>
> [2] Bolya D, et al. "Token merging: Your vit but faster." arXiv preprint arXiv:2210.09461, 2022.
>
> [3] Liang W, et al. "Expediting large-scale vision transformer for dense prediction without fine-tuning." Advances in Neural Information Processing Systems, 2022, 35: 35462-35477.

---

### Official Review · Reviewer_W4sK · 2023-11-07

**Soundness:** 3 good
**Presentation:** 3 good
**Contribution:** 3 good
**Rating:** 6
**Confidence:** 3

**Summary:**

In "Convolution Meets LoRA" the authors introduce a method for parameter-efficient finetuning of the Segment Anything Model (SAM) in specialized domains where it may initially underperform. The approach's effectiveness is demonstrated across a diverse range of datasets and is rigorously compared against a substantial set of baseline methods.

The method, referred to as Conv-LoRA, involves the incorporation of a modified version of low-rank adaptation (LoRA) into SAM's image encoder. Conv-LoRA improves upon LoRA by introducing a convolutional layer at its bottleneck, and includes a Mixture of Experts module to dynamically select the convolutional layer's scale of operation. Furthermore, the authors extend SAM's functionality to address multi-class segmentation tasks without the need for explicit prompts, allowing it to be deployed in an end-to-end setting.

**Strengths:**

This paper aims at solving the important problem of adapting a foundation model for computer vision to new, specialized domains. The authors propose Conv-LoRA - a new method for parameter-efficient fine-tuning (PEFT) of the segment anything model (SAM), which allows the adaptation of SAM to new domains where it may initially underperform. The combination of low rank adaptation (LoRA) with convolutions at various scales is a novel concept and a valuable contribution to the field of parameter-efficient finetuning for vision transformers.

The authors provide a substantial assessment of the quality of their work by showing a small but robust improvement over various baselines in a diverse set of datasets. While the relationship to some prior work needs to be elaborated upon, the paper includes a good overview of current efforts for PEFT. The authors improve the reliability of their results by running most experiments three times, thus reducing the likelihood of spurious effects stemming from random initialization.

The paper is generally well-structured and easy to follow. The authors effectively convey their approach and findings to the reader with only minor clarifications needed (as noted in the reviewer comments).

With the introduction of Conv-LoRA this paper not only expands the applicability of SAM to a wider range of datasets but also introduces a new method of PEFT that could be applicable to Vision Transformers more generally.

**Weaknesses:**

The paper convincingly demonstrates the effectiveness of the PERF method it introduces, but its motivation and explanation for why it works is unintuitive to me and the supporting evidence is insufficient. The authors claim that adding a convolutional layers reintroduces the inductive biases that are helpful in the image domain and hard-coded into convolutional layers. However, in Conv-LoRA, the convolutional layers are not applied to images but on features that do not necessarily adhere to the locality prior by construction. Can the locality prior truly be reintroduced if it might have already been lost, or, should this rather be regarded as a data-efficient method for finetuning that utilizes the learned locality of the early features in the ViT (see e.g. Raghu, Maithra, et al. 2021)? Unless my understanding of this problem is lacking, the explanation for the good performance of Conv-LoRA should be reformulated as a hypothesis.

Similarly, the authors identify "SAM's foreground-background segmentation pretraining" as a weakness, but SAM actually outputs three (prompt-dependent) masks with the idea of allowing the network to identify an object hierarchy (whole, part, sub-part) for ambiguous prompts. To me that seems closely related to multi-class segmentation and requires understanding of the image semantics. I think the paper could be strengthened by elaborating on and providing evidence for this deficiency of SAM.

While the paper includes an overview of other efforts for parameter-efficient finetuning (PERF), a clarification on what sets it apart from other work on PERF of SAM would strengthen their work. Specifically, the authors mention the work of Zhang&Liu 2023 (SAMed), which also uses LoRA to adapt SAM to a new domain (medical images) and from my understanding also repurposes SAM to work for multi-class segmentation in an end-to-end fashion. Considering that Zhang&Liu 2023 is a very recent work and has not been published in a peer-reviewed venue, a direct comparison cannot be expected. However, the authors should revisit their claim in the introduction that Zhang&Liu (2023) fail to address SAM's limitation of not capturing "high-level image semantic information" (point 2) and functioning as a "standalone, end-to-end solution" (point 3). Similarly, the authors cite Shaharabany et al., 2023, who also adapt SAM to work fully automatically (point 3).

In summary, the authors show through their extensive set of experiments that their method is an effective way of performing PEFT of SAM to difficult domains and is therefore a valuable contribution. The concerns above are only with regards to the representation of prior work and the explanation of why Conv-LoRA is effective, not with the soundness of the method itself and the scientific rigor in showing its effectiveness. If the authors can address these concerns in the discussion or, where applicable, with minor edits to the language in the paper, I recommend accepting this work.

Raghu, Maithra, et al. "Do vision transformers see like convolutional neural networks?." Advances in Neural Information Processing Systems 34 (2021): 12116-12128.
Zhang, Kaidong, and Dong Liu. "Customized segment anything model for medical image segmentation." arXiv preprint arXiv:2304.13785 (2023).
Shaharabany, Tal, et al. "AutoSAM: Adapting SAM to Medical Images by Overloading the Prompt Encoder." arXiv preprint arXiv:2306.06370 (2023).

**Questions:**

1. As discussed above, I am not fully convinced of the author's explanation for the improved performance they see with Conv-LoRA compared to LoRA. My understanding is that while ViTs learn about locality in images, this is not a prior that is built into the architecture. What do the authors mean by "SAM's local prior assumption" mentioned in the abstract? How can "image-related inductive biases" be reinjected on top of features that do not have those biases?
2. I don't follow the author's reasoning for why SAM's foreground-background pretraining is insufficient (see above). Can you elaborate on this?
3. My understanding of the cited work Zhang&Liu 2023 is that they also adapt SAM to work without an image-dependent prior and they also extract semantics from the segmentation head. Similarly, Shaharabany, Tal, et al. adapt SAM to work fully automatically. Please make sure to not misrepresent these works in the introduction (see above).
4. I am confused about the author's choices with regards to the scaling for the convolutional layers. What is the interpolation method used? Is it learned? If not, can the larger upsampling factors do much here with the chosen kernel size of 3x3? Adding a row to table 4 with a scaling factor of 8 would provide valuable insights.

## Minor Suggestions for Improvement
5. Can you clarify where exactly in the image encoder ViT the Conv-LoRA bypass is added?
6. In the introduction the authors profess that SAM is underperforming on certain domains. The references supporting this claim can be found in the Related Work section (Tang et al., 2023; Ji et al., 2023; Zhou et al., 2023). I suggest that the authors add these references to the first time the claim is made.
7. The authors identify the adaptation of SAM for end-to-end multi-class segmentation as one of the major contributions of their work. I suggest that the authors add a paragraph to the related work section discussing other efforts to do so if there are any.
8. Figures 8, 9, 10 should have their own labels.
9. Make sure all variable names are defined and used consistently (Figure 4, Appendix A). I wasn't able to follow everything in appendix A. (e.g., where does B come from when looking at eq. 2?)
10. I suggest adding a figure with example images for the medical domain because it is featured prominently in the abstract.
11. Consider spelling out low rank adaptation in the abstract before abbreviating it. This would make the paper more welcoming to readers that are new to the field.
12. I wanna encourage the authors to consider releasing their code. This would not only facilitate further research and collaboration in the field but also help realize the impact of their work by making it easier for users to adapt SAM to new domains.

---

> ### Author Response · Authors · 2023-11-22
> **Replies to Reviewer W4sK: first part**
>
> We sincerely appreciate your thorough review of our paper and the valuable insights you provided. In the following sections, we will provide a detailed response to each of your comments.
>
> **Q1(1): What do the authors mean by "SAM's local prior assumption" mentioned in the abstract?**
>
> SAM’s local prior is grounded in its extensive segmentation pretraining. Through supervised training on a vast dataset encompassing 1 billion high-quality masks and 11 million images, **SAM has honed a robust capability to discern and capture local features within images**. Notably, SAM's encoder retains the ViT architecture, which inherently lacks a dedicated local prior. **However, this deficiency is effectively compensated for by the significant local prior acquired through segmentation pretraining**.
>
> To substantiate SAM's local prior, we conduct an analysis using the mean attention distance as a metric [1] [2]. In **Appendix G (Figure 12)**, our findings reveal that SAM exhibits many heads in the deep blocks with short mean attention distances. This observation indicates SAM's heightened focus on local information during the later stages of the encoder. In contrast, the MAE pretrained ViT[3], representing SAM's initialization, displays consistently long mean attention distances among attention heads in the later stages. Consequently, SAM's segmentation pretraining induces a transformative shift in ViT's attention heads, steering them from a global-oriented to a local-oriented configuration. **This transformation underscores the efficacy of SAM's approach in imbuing the model with a distinctive local prior, enhancing its ability to capture fine-grained details within images.**
>
> -------------------------------------------------------------------------------------------------------------------------------------------------------------
>
> **Q1(2): How can "image-related inductive biases" be reinjected on top of features that do not have those biases?**
>
> **It is crucial to clarify that the features do not entirely lose their "image-related inductive biases"**; rather, they retain certain biases even after processing through the model. Extensive pre-training on large-scale datasets has been demonstrated to equip ViTs with these inductive biases [1] [2]. SAM's large-scale segmentation pretraining further amplifies these inductive biases within the features.
>
> To evaluate the inductive biases of the features, we conduct comprehensive comparisons among randomly initialized ViT, MAE pretrained ViT, and SAM ViT. Two key metrics, mean attention distance, and relative log amplitudes of Fourier transformed features[4], are presented in **Figure 13 in Appendix G**. Notably, as the training progresses from randomly initialized ViT to MAE ViT and ultimately to SAM ViT, we observe a consistent trend: the mean attention distance decreases, while the high-frequency signals in the features increase, indicating a focus on local information.
>
> **It is important to underscore that Conv-LoRA is designed not to reintroduce inductive biases but rather to reinforce existing implicit biases within the features**. Throughout the paper, we deliberately use the term "reinforce" to accurately convey the model's objective, avoiding any implication of reintroduction. This distinction reflects our nuanced approach in ensuring that the model builds upon and strengthens the inherent image-related inductive biases within the features.
>
> -------------------------------------------------------------------------------------------------------------------------------------------------------------
>
> **Q1(3): "data-efficient method for finetuning"**
>
> Regarding your speculation about our method being a data-efficient approach, we consider this to be a complementary explanation for the observed performance improvement. Because convolutional operations naturally contain inductive biases.
>
> We also conduct experiments (**Table 16 in Appendix I**) in the few-shot setting, wherein acquiring data for downstream tasks is challenging, and only a limited number of training samples are available. Our experiments are performed on the Trans10K-v2 dataset, which is used for twelve-class transparent object segmentation. We randomly select the training samples, to meet the settings of 1, 2, 4, 8, and 16 shots (i.e., the number of labeled training examples per class), and the experiments are run for 100 epochs:
>
> | Method    |    Shot    |               mIoU                |
> | --------- | :--------: | :-------------------------------: |
> | LoRA      | 1/2/4/8/16 |   13.32/18.01/22.70/25.89/38.11   |
> | Conv-LoRA | 1/2/4/8/16 | **14.53/19.81/23.05/26.02/38.63** |
>
> The performance improvement of Conv-LoRA compared to LoRA, is particularly pronounced in an extremely low-data setting (e.g., 1-shot). The result demonstrates that **the introduction of inductive bias in Conv-LoRA contributes to improved data efficiency.**

---

> ### Author Response · Authors · 2023-11-22
> **Replies to Reviewer W4sK: second part**
>
> **Q2: Why SAM's foreground-background pre-training is insufficient?**
>
> SAM's foreground-background segmentation pretraining is instrumental in discerning object hierarchies. However, **it is important to recognize that the segmentation within an object, though beneficial for hierarchy identification, differs significantly from the demands of multi-class segmentation**. The latter necessitates a holistic understanding of global semantics and the ability to distinguish objects belonging to distinct classes.
>
> SAM was trained on a dataset that exclusively comprises segmentation masks without explicit semantic information. In theory, to minimize loss, the fundamental objective for its encoder is to project pixels into a metric space where pixels from the same object are in close proximity, while those from distinct objects are distantly positioned. This projection requires an implicit understanding of "objectness", focusing on proximity within an image rather than preserving consistent representations of the same-type object across different images. **This introduces a potential challenge in aligning representations with semantics across diverse images.**
>
> To substantiate this assumption, we conducted a linear probing experiment on the ImageNet-1K dataset, utilizing ViT base models from SAM and MAE. The results presented in **Section 4.2** of our paper reveal a notable discrepancy, with MAE exhibiting significantly higher accuracy compared to SAM (67.7% vs. 54.2%). Further evidence is presented in **Figure 12 of Appendix G**, illustrating that many attention heads near the tail of the ViT exhibit relatively small attention distances after SAM pretraining.
>
> *linear probing experiment*:
>
> | Model |   Acc.   |
> | :---: | :------: |
> |  MAE  | **67.7** |
> |  SAM  |   54.2   |
>
> **This tendency of SAM's pretraining to emphasize information retention for low-level segmentation comes at the expense of losing crucial information essential for high-level classification,** a key requirement for multi-class semantic segmentation. These findings collectively underscore the possible limitations of SAM's foreground-background pretraining in adequately addressing the complexities of multi-class semantic segmentation tasks.
>
> ---------------------------------------------------------------------------------------------------------------------------------------------------
>
> **Q3: Misrepresented works.**
>
> Thank you for bringing attention to this aspect. We acknowledge your observation that both Zhang&Liu and Shaharabany et al. have indeed adapted SAM to function as an end-to-end model. In response to your feedback, we have revised our claims, removing the assertion of SAM being an "end-to-end solution" from our list of major contributions.
>
> Regarding the notion of "capturing high-level semantic information," we wish to emphasize that Zhang&Liu did not explicitly delve into a detailed analysis of this particular limitation of SAM. We want to convey that, irrespective of whether these adaptations are made with an awareness of SAM's this limitation, **our emphasis lies in conducting in-depth analyses (linear probing and attention distance statistics) and offering insights into SAM's constraints**. We believe that such analyses are essential in shedding light on potential areas for improvement in SAM in the future.
>
> ---------------------------------------------------------------------------------------------------------------------------------------------------
>
> **Q4: Choice of Interpolation Method and Larger Upsampling Factors.**
>
> We utilize bilinear interpolation and have added a row with a scaling ratio of 8 in **Table 4** of our paper. The findings indicate that **a larger scaling factor does not consistently lead to improved performance, as the object scale exhibits variability across different datasets**. Hence, we propose to let the model itself learn to select scale dynamically through MoE (Mixture of Experts).
>
> ---------------------------------------------------------------------------------------------------------------------------------------------------
>
> **Q5: Where is Conv-LoRA added in ViT?**
>
> We apply Conv-LoRA to the query, key and value matrices in self-attention layers, the same locations of LoRA.
>
> ---------------------------------------------------------------------------------------------------------------------------------------------------
>
> **Q6: Add more references.**
> We have added the suggested references in the first paragraph of introduction.
>
> ---------------------------------------------------------------------------------------------------------------------------------------------------
>
> **Q7: Add related works.**
> We have added one new paragraph in the related work section introducing other concurrent efforts in finetuning SAM. Following your suggestion, we have removed "end-to-end segmentation" from our major contributions.

---

> ### Author Response · Authors · 2023-11-22
> **Replies to Reviewer W4sK: third part**
>
> **Q8: Missing labels in Figures 8,9, and 10**
>
> Good catch. We have added labels in those figures.
>
> -------------------------------------------------------------------------------------------------------------------------------------------------------------
>
> **Q9: Variable name consistency**
>
> We add the definition of the variable names in the caption of **Figure 4.**
>
> In **Equation 2**, the dimensionality of $x$ is denoted as $B \times C_{in} \times H \times W$, where $B$ represents the batch size, $C_{in}$ denotes the number of input channels, and $H$ and $W$ correspond to the height and width, respectively. As encoder handles the channel dimension, the size of $W_ex$ being $B \times r \times H \times W$, where $W_e \in R^{r \times C_{in}}$. Previously, the batch size dimension was omitted; however, for clarity, we will now explicitly state the size of $x$ in Equation 2.
>
> -------------------------------------------------------------------------------------------------------------------------------------------------------------
>
> **Q10: Add example images from the medical domain**
>
> We have provided some medical image examples in **Figure 6** **in Appendix E**.
>
> -------------------------------------------------------------------------------------------------------------------------------------------------------------
>
> **Q11: Spell out LoRA**
>
> We will update abstract with the full name of LoRA, i.e., low-rank adaptation. As ICLR2024 does not allow to change the abstract during discussion stage, we will modify "LoRA" to "Low Rank Adaptation" later.
>
> -------------------------------------------------------------------------------------------------------------------------------------------------------------
>
> **Q12: Code Release**
>
> Good call. We are cleaning up our codebase and will release it later. We definitely want to contribute to the open-source community and facilitate future research.
>
> -------------------------------------------------------------------------------------------------------------------------------------------------------------
>
> References:
>
> [1] Dosovitskiy A, et al. "An image is worth 16x16 words: Transformers for image recognition at scale." arXiv preprint arXiv:2010.11929, 2020.
>
> [2] Raghu, Maithra, et al. "Do vision transformers see like convolutional neural networks?." Advances in Neural Information Processing Systems 34 (2021): 12116-12128. Zhang, Kaidong, and Dong Liu.
>
> [3] Kaiming He, et al. "Masked autoencoders are scalable vision learners." Proceedings of the IEEE/CVF conference on computer vision and pattern recognition, pp. 16000–16009, 2022.
>
> [4] Park N, et al. "How do vision transformers work?." arXiv preprint arXiv:2202.06709, 2022.

---

### Author Response · Authors · 2023-11-22
**General response to all reviewers.**

We thank all reviewers' efforts in evaluating our work. We are glad that our paper is appreciated as well-structured (Reviewer W4sK, Reviewer LN6g, Reviewer 2Lzf) and having convincing results (Reviewer W4sK, Reviewer 3rwQ, Reviewer LN6g, Reviewer 2Lzf).

We also acknowledge all reviewer’s feedback on our work, and summarize the main revisions below. For details, please check the updated files with all revisions highlighted in blue.

1. We delve deeper into the analysis of 'SAM's local prior assumption' and explore the 'image-related inductive biases' within features (**Appendix G**).
2. We add experiments under low-data regime to assess whether introducing inductive biases to SAM can enhance data efficiency (**Appendix I**).
3. We elaborate on the potential reasons why SAM's foreground-background pre-training is insufficient for multi-class semantic segmentation (**Section 4.2**).
4. We further explain the motivation of introducing MoE (Mixture of Experts) in **Section 3.1**, and conduct more ablation experiment (e.g., MoE-Conv vs. Blocks with various kernel sizes in **Appendix F**) and analysis (**Appendix H**).
5. We also provide computational cost comparison and the results compared to SAM trained from scratch (**Appendix F**).

---

### Meta-Review · Area_Chair_inn7 · 2023-12-09

**Metareview:**

The paper proposes a fine-tuning method for the SAM foundation model, developing a convolutional architecture for LoRA. The reviewers unanimously recommend acceptance. The paper is well written and the method is well evaluated.

**Justification For Why Not Higher Score:**

Reviewers cite a lack of clear understanding and motivation for the proposed technique.

**Justification For Why Not Lower Score:**

Clear empirical evaluation of the proposed method.

---

### Decision · Program_Chairs · 2024-01-16

Accept (poster)